# COURSE CORRECTING KOOPMAN REPRESENTATIONS

**Mahan Fathi**
Google DeepMind, Mila
Université de Montréal

**Clement Gehring**
Mila
Université de Montréal

**Jonathan Pilault**
Mila
Polytechnique Montréal

**David Kanaa**
Mila

**Pierre-Luc Bacon**
CIFAR AI Chair, Mila
Université de Montréal

**Ross Goroshin**
Google DeepMind

## ABSTRACT

Koopman representations aim to learn features of nonlinear dynamical systems (NLDS) which lead to linear dynamics in the latent space. Theoretically, such features can be used to simplify many problems in modeling and control of NLDS. In this work we study autoencoder formulations of this problem, and different ways they can be used to model dynamics, specifically for future state prediction over long horizons. We discover several limitations of predicting future states in the latent space and propose an inference-time mechanism, which we refer to as Periodic Reencoding, for faithfully capturing long term dynamics. We justify this method both analytically and empirically via experiments in low and high dimensional NLDS.

## 1 INTRODUCTION

Recent research has shown a growing interest in learning representations of nonlinear dynamical systems (NLDS) in which the dynamics become linear in the latent space. Linear dynamics in the latent space offer distinct advantages, including the capability to derive closed-form solutions to optimal control problems using LQR solvers (Kalman, 1960). System identification and interpretability are also greatly simplified for linear dynamical systems. From a computational standpoint, advancing a linear system forward can be executed more efficiently by leveraging parallelism (Gu et al., 2022; Smith et al., 2023). An example of this approach is presented in S4 by Gu et al. (2022), a carefully designed deep state-space model (SSM) that takes advantage of the parallelism offered by discrete linear dynamics.

Given the advantages of linear systems, a challenge in the study of dynamics and data-driven modelling resides in extracting global linear representations of nonlinear systems. In this regard, Koopman Theory (Koopman, 1931; Koopman & v. Neumann, 1932) provides a framework in which nonlinear dynamics can be cast into linear ones in a space of measurements, spanned by a basis of characteristic functions, which are the eigenfunctions of Koopman's composition operator. However, identifying such characteristic functions as well as the structure of the linear transition operator can prove very difficult, in general, as they depend heavily on the qualitative dynamical properties of the system under study, necessitating the need to resort to approximate methods.

Recently, several studies have explored the integration of deep learning architectures with Koopman operator theory, primarily in the context of small scale well-behaved problems. Azencot et al. (2020) train a backward-compatible Koopman matrix in conjunction with its original forward counterpart, by assuming reversible dynamics, ensuring stability during training by driving the eigenvalues of the operators to be close to one. Frion et al. (2023) adopt a similar strategy by promoting orthogonality of the Koopman matrix through an auxiliary loss. Lusch et al. (2018a) learn generalized Koopman representations for systems including those exhibiting continuous spectra. Mondal et al. (2023) employ a Koopman-based model in the context of Model-based Reinforcement Learning as a short-horizon (20-step) planner.

In this work, we further explore the use of principles outlined by Koopman theory (Brunton et al., 2022), as a guiding framework to obtain linear representations of NLDS within an autoencoder

framework. We observe that unrolling the linear dynamics in latent space leads to long term drift in trajectories when mapped back to the space of observables (state space). Prior works achieved stable unrolls at inference time by restricting the eigenvalues of the Koopman/linear operator, and training on long sequences. We posit that such requirements, i.e. long training sequences, are unnecessary, especially when modelling a fully observable dynamical system. In this work we uncover two limitations of generating trajectories in the latent space: (i) long horizon trajectories cross, violating uniqueness of solutions of dynamical systems, and (ii) latent space trajectory generation is unable to capture switching dynamics between fixed points (Lan & Mezić, 2013). To overcome these issues, we introduce a simple inference method called *Periodic Reencoding* that produces high accuracy predictions over long horizons. We derive analytic expressions and present a special case example (Appendix D) that provides intuition about why this method better captures long-term dynamics. Finally, we empirically validate our approach on a set of established NLDSs, and also demonstrate these findings on offline reinforcement learning datasets in more complex environments within D4RL (Fu et al., 2020).

## 2 DEEP KOOPMAN AUTOENCODERS

Given a nonlinear dynamical system, Koopman theory attempts to approximate or "explain" nonlinear transitions with a linear dynamical system – refer to Appendix B for a detailed overview of the Koopman theory. For example, Dynamic Mode Decomposition (DMD) (Brunton et al., 2022) tries to approximate the infinite dimensional Koopman operator by fitting discrete transition data collected from a nonlinear dynamical system. More specifically, using an integration scheme in time a discrete nonlinear dynamical system can be obtained from its continuous time counterpart, $x_{t+1} = \mathbf{F}(x_t)$. DMD then solves the following optimization problem over the dataset of all transition pairs, $\mathcal{D} = \{(x_t, x_{t+1}) \mid x_t, x_{t+1} \in \mathbb{R}^d\}$:

$$\mathbf{K}^{\text{DMD}} = \arg\min_{\hat{\mathbf{K}}} \sum_{\mathcal{D}} \|x_{t+1} - \hat{\mathbf{K}}x_t\|_2, \tag{1}$$

where the $\mathbf{K}^{\text{DMD}} \in \mathbb{R}^{d \times d}$ matrix is the finite-dimensional approximation to the linear Koopman operator. Thus DMD treats the state itself as the features, by simply fitting the Koopman matrix directly. Extensions to this approach include "extended" DMD, eDMD (Williams et al., 2015; 2014; Schütte et al., 2016), which augments the state with fixed nonlinear transformations of the state. In the spirit of end-to-end learning, we are interested in data driven approaches that learn nonlinear transformations of the state (Brunton et al., 2022; Lusch et al., 2018b). Koopman features can be learned in an autoencoder setting by minimizing:

$$\min_{\hat{\mathbf{K}}, \phi, \psi} \sum_{\mathcal{D}} \|x_t - \psi(\phi(x_t))\|_2 + \lambda \cdot \|\phi(x_{t+1}) - \hat{\mathbf{K}}\phi(x_t)\|_2 \tag{2}$$

where the encoder $\phi : \mathbb{R}^d \to \mathbb{R}^n$, decoder $\psi : \mathbb{R}^n \to \mathbb{R}^d$ are parameterized function approximators, and $\lambda > 0$ is a scalar. The feature vector $z_t \in \mathbb{R}^n$, representing the Koopman embedding for which the dynamics are linear, obeys the following relations:

$$z_t \approx \phi(x_t), \quad z_{t+1} \approx \mathbf{K}z_t, \quad x_t \approx \psi(z_t) \tag{3}$$

These relations are approximate because, for example, reconstructed states are approximations of the corresponding true states. Once trained, the encoder, decoder, and $\mathbf{K} \in \mathbb{R}^{n \times n}$ comprise a complete model of the dynamical system, potentially capable of performing long-range state prediction over arbitrarily long time horizons.

Though other feature learning methods, such as contrastive learning, are possible (Lyu et al., 2023), we study the autoencoder formulation because it is more pervasive in the literature. Furthermore, the learned decoders from this formulation allow for solving the control problems in latent space and mapping control signals back to phase space. This is non-trivial using contrastive learning which only trains an encoder.

## 3 METHOD

### 3.1 TRAINING SEQUENCE

In this section we formally outline the training objective for sequential data. We start by using the continuous parameterization of the Koopman dynamics, for *controlled systems*. An autonomous controlled system is described by $\dot{x} = f(x, u)$, where $u$ is an exogenous control input. Assuming that a bounded Koopman matrix, $K \in \mathbb{R}^{n \times n}$, can be approximated for measurements $z = \phi(x) \in \mathbb{R}^n$, the linear dynamics are then prescribed by:

$$\frac{\mathrm{d}}{\mathrm{d}t}\phi(x) = K\phi(x) + L\omega(u), \tag{4}$$

where $L \in \mathbb{R}^{n \times m}$ represents the controlled latent dynamics for external coded input $\upsilon = \omega(u) \in \mathbb{R}^m$. We optimize the objective by taking gradient steps directly over the continuous parameterization of the Koopman dynamics, i.e. $K$ and $L$, which we discretize via the bilinear method (Tustin, 1947), over a timestep of $\delta$:

$$z_{t+1} = \mathbf{K}z_t + \mathbf{L}\upsilon_t, \tag{5}$$

$$\text{where} \quad \mathbf{K} = \left(I - \frac{\delta}{2}K\right)^{-1}\left(I + \frac{\delta}{2}K\right) \quad \text{and} \quad \mathbf{L} = \left(I - \frac{\delta}{2}K\right)^{-1}\delta L. \tag{6}$$

We treat $\delta$ as a trainable variable as well, and assume that samples are uniformly distributed in time. The Koopman Autoencoder architecture consists of the following trainable components, (i) the latent dynamics $K$, $L$ and $\delta$ (ii) the state encoder $\phi$ (iii) the action encoder $\omega$ (iv) and the state decoder $\psi$. The training data consists of an initial state, $x_t$, and a sequence of following actions and states, i.e. $(u_t, x_{t+1}, \cdots, u_{t+T-1}, x_{t+T})$ of length $T$. The model takes in the initial state and the sequence of actions as input and is tasked to predict the sequence of future states. To respect the Koopman dynamics, i.e. ensuring that $\mathbf{K}$ and $\mathbf{L}$ are the only means for advancing the dynamics, we minimize the "Aligment", "Reconstruction", and "Prediction" losses that prevent trivial solutions.

$$\mathcal{L}_{\text{Align}} = \sum_{i=1}^{T}\|\hat{z}_{t+i} - \phi(x_{t+i})\|_2 \quad \mathcal{L}_{\text{Reconst}} = \sum_{i=0}^{T}\|x_{t+i} - \psi(z_{t+i})\|_2 \quad \mathcal{L}_{\text{Pred}} = \sum_{i=1}^{T}\|x_{t+i} - \psi(\hat{z}_{t+i})\|_2 \tag{7}$$

In the above equations, $\hat{z}_t$ denotes the resultant latent code after one or more applications of Koopman dynamics, while $z_t$ represents the resultant latent state immediately after encoding (see Figure 1).

### 3.2 TRAJECTORY GENERATION

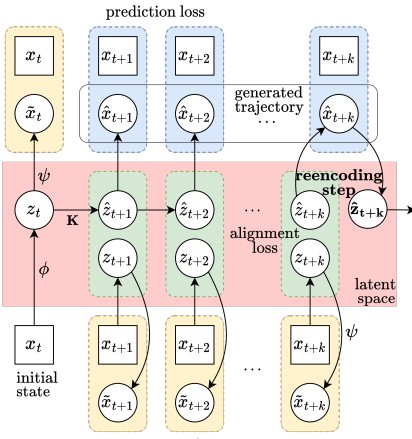

Figure 1: Course Correcting Koopman Autoencoder Unrolls via Periodic Reencoding. Unrolls are generated by first encoding the initial condition, then linearly advancing the dynamics in latent space, and finally decoding the trajectory back to the original space. Values enclosed within squares represent ground truth, while those enclosed within circles are inferred by the model. The objective consists of an alignment loss (shown in green), a reconstruction loss (in yellow), a prediction loss (in blue). We propose Periodic Reencoding; at every $k$ steps in the figure. The latent state, $\hat{\mathbf{z}}_{\mathbf{t+k}}$, is decoded and subsequently reencoded. Because the encoder/decoder are not exact inverses, the reencoded feature vector is different from the original. Control inputs are omitted for simplicity.

The quality of a model of a dynamical system can be assessed by generating trajectories in the state space of the original dynamical system, starting from some initial condition. There are two methods for generating trajectories.

**Without Reencoding.** This method generates the entire trajectory in the latent space and only uses the encoder to obtain the initial condition. The decoder $\psi$ is applied to every point on generated trajectory to obtain the corresponding *curve* in state space. More specifically, given:

$$\dot{z} = Kz|_{z_0 = \phi(x_0)} \quad \text{and} \quad x = \psi(z),$$

we can write the explicit solution in state space as :

$$x(t) = \psi(e^{Kt}\phi(x_0)). \tag{8}$$

It is the solution to a linear dynamical system, given by the matrix exponential, in latent space which is then mapped to the original state space using $\psi$. Under this setting, the relationship between $z-$space and $x-$space is established by mapping *trajectories* generated in the $z-$space via linear dynamics, to *curves* in the $x-$space.

In the standard autoencoder setup, the mappings between the original and latent spaces are not strictly one-to-one. This characteristic becomes especially significant when the dimensionality of the latent space is substantially larger than that of the original space, i.e. $n \gg d$, as is typical in Koopman autoencoders – the inverse function theorem requires that $n = d$. Because of this, the curves generated without reencoding could potentially intersect with themselves, and therefore won't faithfuly capture the characteristics of a *trajectory*, generated by a dynamical system. This directly arises from the existence of multiple points in $z-$space that lie on the same trajectory generated by the linear system, that are mapped to the same point in $x-$space, due to lack of injectivity in the decoder, $\psi$. This is clearly illustrated in Figure 4 (a) where phase lines intersect. Intersecting phase lines are impossible to generate with continuous dynamical systems because they imply that multiple solutions exist corresponding to the unique initial condition given by the cross-over point.

**With Reencoding.** This method uses the encoder/decoder to iteratively generate a trajectory in state-space. The encoder, Koopman operator, and decoder effectively define a dynamical system in the state space, $x$. More specifically, given:

$$\dot{z} = Kz|_{z_0=\phi(x_0)}, \quad z = \phi(x), \quad \text{and} \quad x = \psi(z),$$

we can express the dynamical system that implicitly defines the solution in state space as:

$$\dot{x} = J_\psi\big(\phi(x)\big)K\phi(x) \tag{9}$$

Where $J_\psi$ is the Jacobian of $\psi$. When generating trajectories without reencoding, linearity in the latent space is assumed to hold for all times, i.e. globally. In contrast, when generating trajectories with reencoding, the linearity property is assumed to hold only locally. Equation 9 gives rise to a *dynamical system with feedback*, thus the relationship between $z-$space and $x-$space is governed by point-wise mapping of dynamics. Appendix A shows extensions to *any* latent state dynamics.

In theory, the trajectories generated using this method can faithfully capture the dynamics and will not produce invalid trajectories that cross, however, reencoding at every step poses two significant drawbacks: (i) repeated applications of the encoder can potentially accumulate errors much faster than repeated applications of $\mathbf{K}$, as seen in Figure 4 (b), and (ii) it is not computationally efficient because unrolling must be performed sequentially due to presence of nonlinear operations at every step, unlike the parallelizability of linear operations.

We introduce ***Periodic Reencoding***, a technique spanning the middle ground between trajectory generations with and without reencoding. When generating trajectories with periodic reencoding, we decode and reencode periodically, in continuous time at $\Delta t$ intervals (Figure 2), and in discrete time every $k$ steps (Figure 1), treating $\Delta t$ or $k$ as hyper-parameters. Given that the encoder and decoder are not perfect inverses of one another, the output of the reencoding step is going to be different from the original point. In other terms, the mapping between the original and latent space lacks *bijectivity*.

By Periodic Reencoding, we expand and harness the applicability of local linear Koopman dynamics around the initial state all the while mitigating the accumulation of encoding errors. We observe that periodically applying reencoding is an effective way to mitigate the drift accumulated by generating trajectories in the latent space. We term this property **"course correction."** Using our method, we are able to generate stable, plausible, and accurate unrolls over extended horizons, while remaining computationally efficient. In Section 4, we empirically establish the effectiveness of periodic reencoding for stable and accurate long-range predictions, in contrast to unrolls without reencoding or reencoding at every step. Furthermore, we demonstrate that periodic reencoding can be beneficially integrated into the training process to achieve further improvements. Lan & Mezić (2013) showed that NLDS with multiple fixed points can only be linearized within the basin of attraction of each fixed point. This reveals another limitation of unrolls without reencoding – they cannot capture the switching dynamics between multiple fixed points (see the example in Appendix D). It is important to make the distinction between reencoding and "teacher forcing" (Bengio et al., 2015). Teacher forcing periodically uses ground truth during training to avoid error accumulation. Reencoding never uses ground truth data.

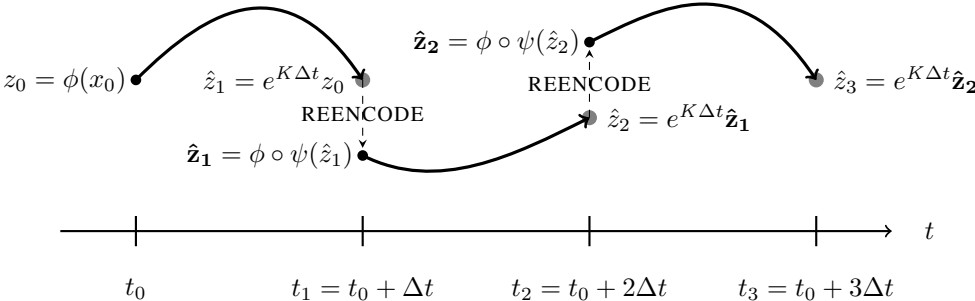

Figure 2: Periodic Reencoding in continuous time for trajectories in latent space. In this figure, the unrolled latent before and after reencoding are denoted as $\hat{z}_i$ and $\hat{\mathbf{z}}_\mathbf{i}$, respectively.

## 4 RESULTS

We empirically evaluated our proposed approach on a number of highly nonlinear environments with varying dimensionality. We begin by modelling the forward dynamics of well known, nonlinear, low dimensional dynamical systems. We further extend the results to more practical, higher dimensional, robotic environments implemented in MuJoCo. We use the D4RL dataset by Fu et al. (2020) to train our Koopman autoencoder. Lastly we employ our proposed approach as an open-loop controller for locomotion tasks in D4RL.

### 4.1 DYNAMICAL SYSTEMS

We use well-established dynamical systems as benchmarks for forward dynamics modeling. Despite their low dimensionality, these systems display interesting nonlinear dynamics, including multiple fixed points. Nonetheless, their low dimensionality enables the generation of informative visual representations in 2D/3D phase plots. We briefly review the environments used in this section.

**Parabolic Attractor**, adopted from Tu et al. (2014); Brunton et al. (2016), is a dynamical system with a single fixed point at the origin, known for its closed-form Koopman embedding solution. Governed by the following equations:

$$\dot{x}_1 = \mu x_1, \quad \dot{x}_2 = \lambda(x_2 - x_1^2), \tag{10}$$

the system admits a solution that is asymptotically attracted to the parabolic manifold, given by $x_2 = x_1^2$, for $\lambda < \mu < 0$. The Koopman embedding, $z$, that adheres to globally linear dynamics, can be coded by augmenting the state with the additional nonlinear measurement of $z_3 = x_1^2$:

$$\dot{z} = \begin{bmatrix} \dot{z}_1 \\ \dot{z}_2 \\ \dot{z}_3 \end{bmatrix} = \begin{bmatrix} \mu & 0 & 0 \\ 0 & \lambda & -\lambda \\ 0 & 0 & 2\mu \end{bmatrix} \begin{bmatrix} z_1 \\ z_2 \\ z_3 \end{bmatrix} \quad \text{for} \quad \begin{bmatrix} z_1 \\ z_2 \\ z_3 \end{bmatrix} = \begin{bmatrix} x_1 \\ x_2 \\ x_1^2 \end{bmatrix} \tag{11}$$

We set $\lambda = -1.0$ and $\mu = -0.1$ and we sample initial conditions uniformly from $x_1, x_2 \in [-1, 1]$.

**Duffing Oscillator** follows a nonlinear second order differential equation $\ddot{x} = x - x^3$, which represents a model for the motion of a damped and force-driven particle. This particular instance admits two center points at $(x, \dot{x}) = (\pm 1, 0)$, and an unstable fixed point at the origin, $(x, \dot{x}) = (0, 0)$. Initial conditions are sampled uniformly from $x_1 \in [-2, 2]$ and $x_2 \in [-1, 1]$.

**Lotka-Volterra** represents the population evolution of biological systems, based on a predator-prey interactions, by the following equations:

$$\dot{x}_1 = \alpha x_1 - \beta x_1 x_2, \quad \dot{x}_2 = \delta x_1 x_2 - \gamma x_2. \tag{12}$$

The system is known for its abrupt switch in population growth and admits two fixed points, one at the origin (extinction), and a center point at $(x_1, x_2) = (\gamma/\delta, \alpha/\beta)$. We set $\alpha = \beta = \gamma = \delta = 0.2$ and uniformly sample initial conditions from $x_1, x_2 \in [0.02, 3.0]$.

**Pendulum** represents a freely swinging pole. The initial conditions indicate the states from which the pole is released, deviating slightly from the inverted position by $\pm 10°$. The state consists of the angle and the angular velocity and we report errors in radians.

**Lorenz System** is a chaotic dynamical system (Lorenz, 1963). It features equilibrium points, some stable and some unstable, and is renowned for the "butterfly effect" arising from its sensitivity to initial conditions. The governing equations are as follows:

$$\dot{x}_1 = \sigma(x_2 - x_1), \quad \dot{x}_2 = x_1(\rho - x_3) - x_2, \quad \dot{x}_3 = x_1 x_2 - \beta x_3 \quad (13)$$

We use the original parameters from Lorenz'63 system. Initial conditions are generated by perturbing the point $(0, 1, 1.05)$ with Gaussian-distributed noise having a standard deviation of 1.

To demonstrate data-efficiency, we applied our proposed Koopman Autoencoder to modest datasets of trajectories gathered from each of the aforementioned dynamical systems, comprising 100 trajectories for Lorenz systems and 50 trajectories for non-chaotic ones. For all dynamical systems except Lorenz'63, the model is trained using the first 500 steps of the trajectories. Nevertheless, during inference, we unroll the models for up to 1000 steps, demonstrating the capability of our approach to accurately capture the underlying dynamics and generalize to unseen regions of the state space. We employed timesteps of 0.01 for forward integration in all environments, with the exception of Lorenz, for which we used a timestep of 0.02. We utilize embedding size of 128 for the dynamical systems.

Results are presented in Table 1. We conducted experiments involving both linear and nonlinear decoders, linear and nonlinear latent dynamics, as well as experiments with and without periodic reencoding. In the nonlinear latent dynamics setting we use an MLP, instead of $K$, to drive the latent state forward. We maintain consistent encoder and decoder capacities across different models applied to the same environment. We train a standard MLP for single step dynamics prediction as baseline, with capacity roughly equivalent to that of the encoders.

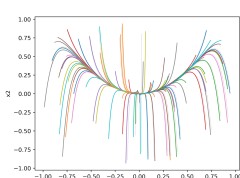

(a) Without reencoding.

Periodic reencoding consistently improves the accuracy of the predictions. These improvements also extend to the setting where we permit nonlinear dynamics in the latent space, even though our method was motivated by learning linear latent dynamics. The only exception is the Parabolic Attractor environment (see Figure 3a). This is expected because this system admits a simple feature transformation that achieves *globally* linear dynamics. The representation can be decoded, with a linear decoder (Equation 11) by simply selecting the first two elements, without the need for course correction. Furthermore, it should be noted that the best prediction results use a linear decoder, rather than a nonlinear one. This observation holds true, regardless of the type of dynamics assumed in the latent space or the specific environment. The results also demonstrate that Koopman autoencoders with linear dynamics and decoders, unrolled using periodic reencoding, consistently outperform widely used nonlinear dynamics models.

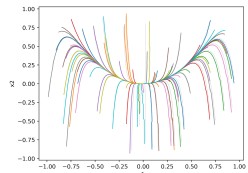

(b) With reencoding.

Figure 3:
Parabolic Attractor.

## 4.2 D4RL: STATE PREDICTION

In this section, we train our proposed Koopman Autoencoder on continuous control tasks from D4RL benchmark by Fu et al. (2020). Again, the goal is to predict the future states over long time horizons. We choose a number of locomotion tasks as our main testbed, namely the `Hopper-v2`, `HalfCheetah-v2`, and `Walker2d-v2` environments. The curated datasets are generated by driving the simulated robots forward from a stationary position, using policies with varying degrees of optimality, i.e. `expert`, `medium-expert`, `medium`, and `medium-replay`. Each individual dataset comprises 1 million transitions, organized as trajectories with a maximum length of 1,000 steps. We use 80% of the trajectories for training and evaluate on the remainder 20%, consisting of trajectories of length 300. Table 2 provides a breakdown of the MSE over a 300-step horizon.

The D4RL benchmark differs from the dynamical systems discussed in the previous section in three fundamental ways: the dynamics are notably more challenging due to the presence of collisions, the dimensionality is higher, and the systems are subject to control inputs. Here we use the extended version of the Koopman autoencoder model designed for controlled systems (see Section 3.1). We use embeddings sizes of 512 and 256 for the state and action encodings, respectively. For training, we solely rely on the observations and avoid incorporating additional training signals, such as rewards.

| MODEL | KOOPMAN (LINEAR LATENT DYNAMICS) | | | | NONLINEAR LATENT DYNAMICS | | | | MLP |
|---|---|---|---|---|---|---|---|---|---|
| DECODER TYPE | LINEAR | | NONLINEAR | | LINEAR | | NONLINEAR | | - |
| PERIODIC REENC. (✗, ✓) | ✗ | ✓ | ✗ | ✓ | ✗ | ✓ | ✗ | ✓ | - |
| ENVIRONMENT | *MSE over **100** steps* | | | | | | | | |
| Parabolic Attractor | **0.0205** | **0.0292** | 0.1465 | 0.0758 | 0.0739 | 0.0496 | 0.0727 | 0.0547 | 0.2674 |
| Pendulum | 0.0512 | 0.0042 | 0.0648 | 0.0181 | 0.0288 | **0.0025** | 0.0242 | 0.0034 | 0.7442 |
| Duffing Oscillator | 0.1152 | 0.0112 | 0.1512 | 0.0512 | 0.0450 | **0.0022** | 0.0450 | 0.0112 | 0.4050 |
| Lotka-Volterra | 0.0113 | 0.0072 | 0.0145 | 0.0098 | 0.0128 | **0.0040** | 0.0112 | 0.0060 | 1.4450 |
| Lorenz'63 | ✗ | 11.162 | ✗ | 12.569 | 18.985 | **7.265** | 19.051 | **7.110** | 88.565 |
| ENVIRONMENT | *MSE over **1000** steps* | | | | | | | | |
| Pendulum | 9.2021 | 0.1818 | 14.5800 | 0.6612 | 10.7184 | **0.0841** | 10.5832 | 0.2964 | 55.281 |
| Duffing Oscillator | 20.5440 | 1.0658 | 15.0701 | 2.5312 | 5.7851 | **0.5725** | 9.2451 | 0.93845 | 22.445 |
| Lotka-Volterra | 1.6292 | 0.3961 | 1.6203 | 0.2812 | 0.7261 | **0.2888** | 0.7281 | 0.4324 | 83.205 |
| Lorenz'63 | ✗ | 78.980 | ✗ | 64.838 | ✗ | **59.262** | ✗ | **54.793** | 133.509 |

Table 1: Mean Squared Error of state prediction for a number of dynamical systems. The entries in the table are scaled up by a factor of $100\times$, except for Lorenz system. Evaluation is done via sampling unseen initial points from the dynamical system. The cross marks and check marks in the PERIODIC REENC. rows indicate, respectively, absence and presence of periodic reencoding mechanism during inference time. When enabled, the errors are reported by searching over reencoding periods of $(10, 25, 50, 100)$. The cross marks in place of table entries indicate exploded values. Underlined values denote best performance for Koopman-based models, where bold numbers represent best performance across all models. We exclude the Parabolic Attractor environment when reporting errors over the 1000-step horizon since trajectories would almost perfectly merge onto the parabolic manifold and reach the origin in less time.

Similar to the previous section, a multi-step MLP is trained as a baseline. To ensure a fair comparison, we set the size of the MLP to be equal to the combined size of the Koopman encoder and decoder. In our implementation, this equivalent MLP is implemented by *setting the reencoding period to 1 during training* and deactivating the loss terms associated with the Koopman autoencoder, specifically the "alignment" and "reconstruction" losses, *optimizing only the "prediction" loss* as the objective (see Figure 1). Refer to Appendix C for details and specifics.

We utilize training sequences with a length of 100. The results presented in Table 2 demonstrate that through periodic reencoding, we can consistently unroll our model over extended horizons, surpassing the training length (in this case, over a 300-step horizon). Furthermore, the use of a nonlinear decoder proves to be crucial for accurate prediction. In our experience, training the MLP becomes unstable for horizons longer than 10 steps, and its performance is significantly inferior. However, it fulfills its role as a baseline.

| MODEL | | KOOPMAN AUTOENCODER | | | | MLP |
|---|---|---|---|---|---|---|
| DECODER TYPE | | LINEAR | | NONLINEAR | | - |
| PERIODIC REENCODING (✗, ✓) | | ✗ | ✓ | ✗ | ✓ | - |
| Environment | Dataset | *MSE over **300** steps* | | | | |
| Hopper[v2] | expert | $0.250 \pm 0.05$ | $0.079 \pm 0.03$ | $0.353 \pm 0.05$ | $\mathbf{0.012 \pm 0.00}$ | 0.561 |
| | medium-expert | $0.486 \pm 0.07$ | $0.102 \pm 0.04$ | $0.719 \pm 0.10$ | $\mathbf{0.015 \pm 0.00}$ | 0.592 |
| | medium | $0.624 \pm 0.05$ | $0.102 \pm 0.03$ | $0.889 \pm 0.08$ | $\mathbf{0.008 \pm 0.00}$ | 0.533 |
| | full-replay | $1.052 \pm 0.12$ | $0.570 \pm 0.11$ | $2.254 \pm 0.14$ | $\mathbf{0.158 \pm 0.01}$ | 0.815 |
| | medium-replay | $1.190 \pm 0.22$ | $0.776 \pm 0.18$ | ✗ | $\mathbf{0.296 \pm 0.07}$ | 0.936 |
| HalfCheetah[v2] | expert | $0.645 \pm 0.05$ | $0.602 \pm 0.09$ | $0.622 \pm 0.07$ | $\mathbf{0.227 \pm 0.03}$ | 1.359 |
| | medium-expert | $1.141 \pm 0.05$ | $1.076 \pm 0.08$ | $0.813 \pm 0.12$ | $\mathbf{0.391 \pm 0.07}$ | 1.481 |
| | medium | $1.362 \pm 0.04$ | $1.456 \pm 0.08$ | $1.816 \pm 0.16$ | $\mathbf{0.809 \pm 0.09}$ | 1.861 |
| | full-replay | $1.262 \pm 0.10$ | $1.252 \pm 0.17$ | $1.668 \pm 0.17$ | $\mathbf{0.816 \pm 0.14}$ | 1.994 |
| Walker2d[v2] | expert | $0.364 \pm 0.07$ | $0.302 \pm 0.08$ | $0.544 \pm 0.05$ | $\mathbf{0.072 \pm 0.03}$ | 0.285 |
| | medium-expert | $0.755 \pm 0.02$ | $0.602 \pm 0.08$ | $0.796 \pm 0.10$ | $\mathbf{0.198 \pm 0.08}$ | 1.295 |
| | medium | $0.825 \pm 0.08$ | $0.718 \pm 0.11$ | $1.822 \pm 0.18$ | $\mathbf{0.404 \pm 0.13}$ | 0.821 |
| | full-replay | $1.676 \pm 0.19$ | $1.413 \pm 0.13$ | ✗ | $\mathbf{0.867 \pm 0.15}$ | 1.291 |
| | medium-replay | ✗ | $2.379 \pm 0.20$ | ✗ | $\mathbf{2.077 \pm 0.26}$ | 1.917 |

Table 2: Mean Squared Error of state prediction for D4RL robotic locomotion tasks. Evaluation is done under a held-out set of trajectories. The cross marks in place of table entries indicate exploded or almost exploded values (large errors).

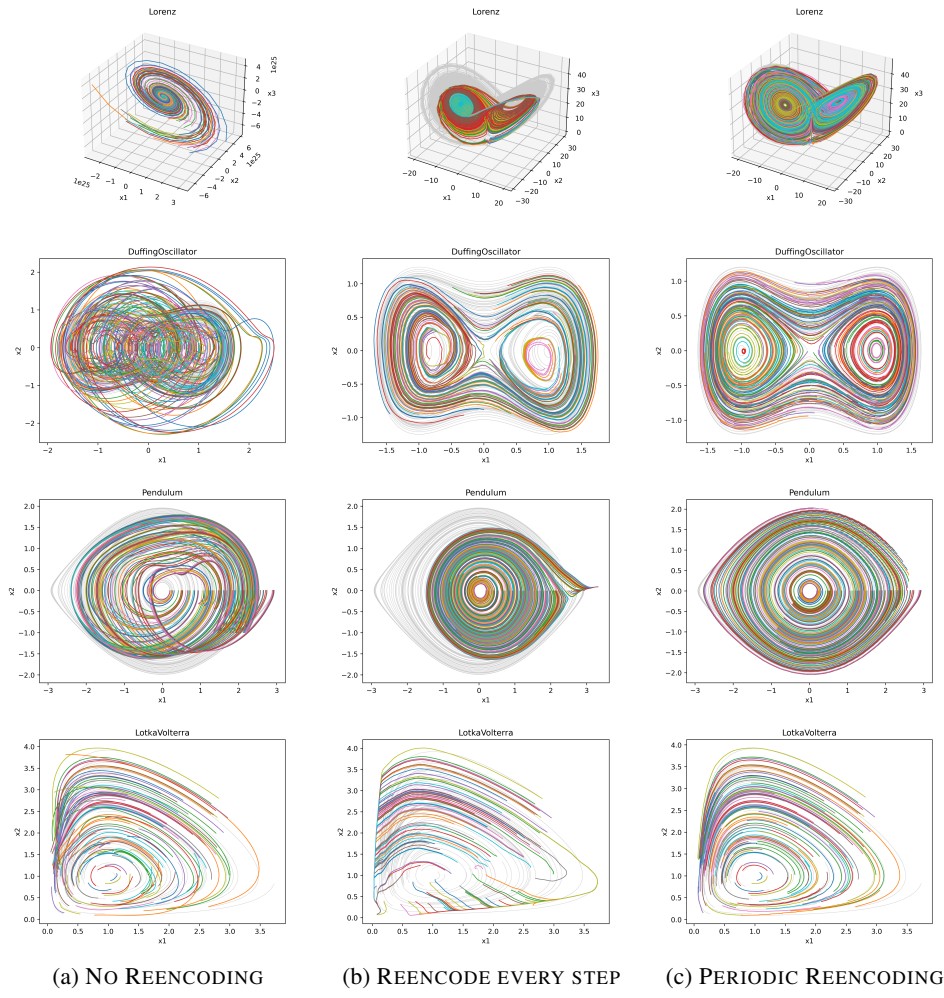

(a) NO REENCODING      (b) REENCODE EVERY STEP      (c) PERIODIC REENCODING

Figure 4: Phase portraits of the trained Koopman autoencoders (w/linear decoder) for unrolls of 1000 steps. We use different different reencoding schemes for unrolling the same model, namely (a) w/o reencoding, (b) reencoding at every step, and (c) periodic reencoding (our proposed approach). The grey lines in the background represent ground truth phase portraits.

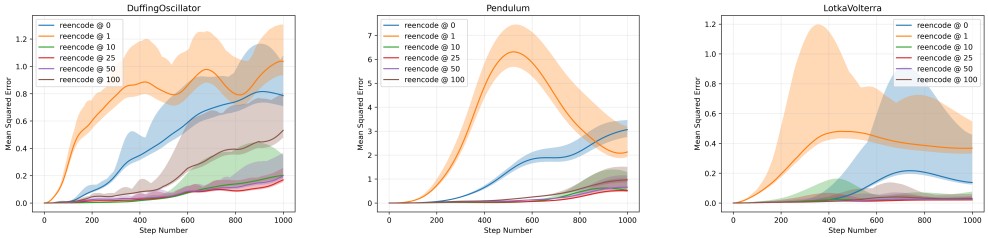

Figure 5: Mean Squared Error (MSE) over the unrolling horizon, under varying reencoding schemes. We use 100 freshly sampled (unseen) initial points for evaluation. `reencode @ 0` legend indicates unrolling without ever reencoding. We observe that Periodic Reencoding robustly improves the quality of long-range modeling of dynamics.

## 4.3 D4RL: SEMI-OPEN-LOOP CONTROL

In this section, we utilize our proposed model as an open-loop controller for locomotion tasks within the D4RL framework to showcase long-term stability, generalization capability, and prediction quality. This approach allows us to present another informative metric, the total reward achieved by a semi-open-loop controller (with sparse feedback). In this context, the model is trained using transitions generated by an optimally trained policy, specifically the `expert` datasets.

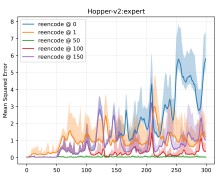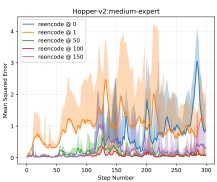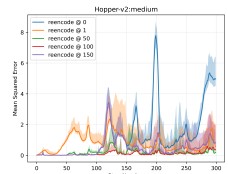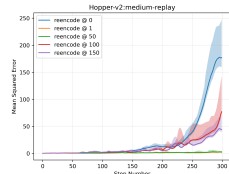

Figure 6: Mean Squared Error (MSE) for Hopper$^{v2}$ environment over the unrolling horizon of 300 steps, under varying reencoding schemes. Periodic Reencoding generates more accurate trajectories compared to no reencoding and every-step reencoding schemes. Error plots for other environments can be found in Appendix F.

The objective is to produce the sequence of states and actions that follow the initial state which is provided as input to the model. We train a Koopman autoencoder, jointly with an independent decoder head trained using a behaviour cloning loss to output actions, which receives the encoded states as input. We assess the quality of the predictions by evaluating the optimality of the replicated "expert" behavior. This is done by executing the generated sequence of actions in the environments and recording the total reward achieved. To underscore the stability of our approach, *we ensure that the model plans the motion of the robot for the next 100 steps*, by generating a sequence of actions before receiving any sort of feedback form the environment. By removing feedback at test time, i.e. by providing the ground truth state as input to the model only every 100 steps, we demonstrate the model's capability to generate stable motion plans, particularly when periodic reencoding is utilized.

| MODEL | | KOOPMAN AUTOENCODER | | MLP | BC |
|---|---|---|---|---|---|
| STATE FEEDBACK (✗, ✓) | | ✗ | ✗ | ✗ | ✓ |
| PERIODIC REENCODING (✗, ✓) | | ✗ | ✓ | - | - |
| Environment | Dataset | *total reward until termination* | | | |
| Hopper$^{v2}$ | expert | $18.40 \pm 6.2$ | $\mathbf{53.5 \pm 14.5}$ | $18.9 \pm 5.9$ | $96.05 \pm 4.3$ |
| HalfCheetah$^{v2}$ | expert | $15.06 \pm 5.3$ | $\mathbf{64.2 \pm 12.9}$ | $19.5 \pm 7.9$ | $82.91 \pm 5.9$ |
| Walker2d$^{v2}$ | expert | $18.46 \pm 8.1$ | $\mathbf{61.9 \pm 14.2}$ | $11.4 \pm 2.5$ | $98.73 \pm 1.5$ |

Table 3: Semi-open-loop reward achieved. The total rewards are normalized according to D4RL random and expert scores. The Koopman Autoencoder uses a nonlinear decoder.

Table 3 provides the results for the designed open-loop control problem. As baseline, we train an MLP that takes the state as input and outputs the next state along with the optimal action. The MLP can then be unrolled autoregressively at test time similarly in an open-loop fashion. Furthermore, for comparison, we train a standard behavior cloning (BC) model that we run *without* state obfuscation, and allow to observe state at every step. This is the upper bound of performance of the policy run with state obfuscation.

We demonstrate that our approach is capable of making sufficiently accurate predictions well into the future when the state is withheld from the model over 100-step horizons. This is evident from the agent's ability to sustain its motion without falling, relying solely on preplanned motion.

## 5 CONCLUSION

Our study of Koopman autoencoders for modeling nonlinear dynamical systems lead us to explore various inference schemes for generating predictions over long horizons. We showed that generating trajectories exclusively in the latent space presents two potential difficulties: (i) the inability to capture switching behaviour between multiple fixed points, and (ii) violation of the existence and uniqueness theorem of initial value problems. We showed, through theory and experiment, that trajectories generated with reencoding do not suffer from these limitations. Finally we introduced periodic reencoding as a method that bridges the gap between no reencoding and reencoding at every step, and achieves the best results in practice.

## ACKNOWLEDGMENTS

We would like to thank Daniel Worrall for careful review of the manuscript and useful feedback. We would also thank Peter Battaglia and Hugo Larochelle for their prompt approval of the work for publication.

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

## A   IMPLICATIONS FOR ALL LATENT MODELS OF DYNAMICAL SYSTEMS

Equations 8 and 9 can be generalized beyond linear latent dynamics to *any deterministic latent space dynamics*. Simply let $K$ be a function of $z$ instead of a constant matrix giving rise to latent space dynamics of $\dot{z} = K(z)$, with solution $z(t) = h(t, \phi(x_0))$. Without reencoding the solution in state space becomes:

$$x(t) = \psi\left(h(t, \phi(x_0))\right). \tag{14}$$

Note that because $h$ is the solution to a dynamical system in $z$, its phase lines would never cross in latent space. However, there is no such guarantee in $x$–space for arbitrary $\psi$. Reecnoding gives rise to the following dynamical system:

$$\dot{x} = J_\psi\big(\phi(x)\big) K\big(\phi(x)\big), \tag{15}$$

where $K$ is now simply a function, taking $z = \phi(x)$ as input. Naturally, the solution would satisfy the uniqueness theorem of initial value problems, i.e. phase lines would not cross. These findings are empirically validated to generalize to the nonlinear case by learning a nonlinear latent dynamics (parameterized by an MLP), see the 'NonLinear Latent Dynamics' column in Table 1.

## B   KOOPMAN THEORY OVERVIEW

Let us consider an autonomous dynamical system defined over an open set $\mathfrak{D}$ of $\mathbb{R}^n$ by a system of first order differential equations:

$$\dot{\mathbf{x}} = \mathbf{f}(\mathbf{x}) \tag{16}$$

This system induces a flow $\Phi : \mathfrak{D} \times \mathbb{R} \times \mathbb{R} \to \mathfrak{D}$ which, for a given time $t$, an observed value of the state of the system $\mathbf{x}_t$ and a duration $\Delta t$, maps to the state of the system evolved by said duration

$$\Phi_{t \to t+\Delta t}(\mathbf{x}_t) = \mathbf{x}_t + \int_t^{t+\Delta t} \mathbf{f}(\mathbf{x}_\tau)\, \mathrm{d}\tau = \mathbf{x}_{t+\Delta t} \tag{17}$$

One of the challenges in modern dynamical system theory revolves around establishing transformations which maps such nonlinear dynamics into linear ones, as there exist myriad of tools and frameworks facilitating the study of linear systems, in contrast to their nonlinear counterparts.

In his seminal work Koopman (1931) proposed an alternative perspective of general nonlinear dynamics in which he considered *measurement functions* $\varphi : \mathfrak{D} \to \mathbb{R}$ s.t. $\varphi \in \mathfrak{L}^2$, the set of square integrable functions w.r.t. Lebesgue's measure—*which constitutes a Hilbert space whose metric is determined by the inner product* $(\varphi, \psi) \mapsto \int_{\mathfrak{D}} \varphi(x)\psi^*(x)\mathrm{d}\mu(x)$. A *measurement function* maps the actual state of the system to the measurement or observable data to which one would have access. He introduced an operator $\mathcal{K}$ acting upon the set of measurement functions and which advances the measurement such that

$$\forall\, \varphi \in \mathfrak{L}^2(\mathfrak{D}),\ \forall \mathbf{x} \in \mathfrak{D},\ \forall t, s \in \mathbb{R}\ \text{s.t.}\ t > s,\ \mathcal{K}_{s \to t}(\varphi)(\mathbf{x}) = \varphi\left(\Phi_{s \to t}(\mathbf{x})\right) \tag{18}$$

It is linear by definition and the family of operator $\{\mathcal{K}_{0 \to t}\}_{t \in \mathbb{R}}$ constitute a one-parameter group in Hilbert space, which admits as infinitesimal generator the *Lie derivative operator* $\mathcal{T}$ such that

$$
\begin{aligned}
\mathcal{L}(\varphi)(\mathbf{x}(t)) &= \lim_{\tau \to 0} \frac{\mathcal{K}_{t \to t+\tau}(\varphi)(\mathbf{x}(t)) - \varphi(\mathbf{x}(t))}{\tau} \\
&= \lim_{\tau \to 0} \frac{\varphi(\mathbf{x}(t+\tau)) - \varphi(\mathbf{x}(t))}{\tau} = \frac{\mathrm{d}}{\mathrm{d}t}\left\{\varphi(\mathbf{x}(t))\right\} = (\nabla \varphi)(\mathbf{x}(t)) \cdot \mathbf{f}(\mathbf{x}(t))
\end{aligned}
\tag{19}
$$

The resulting linear differential equation on measurement functions is of little use in practical settings, however, as it is of infinite dimension. To circumvent this, it is customary to resort to orthogonal decomposition of the Hilbert space of measurement functions $\mathfrak{L}^2$ into subspaces invariant by application of the Lie (and thus Koopman) operator (Brunton et al., 2016; Morton et al., 2018).

$$\mathfrak{L}^2 = \bigoplus_{k=1}^{\infty} \mathfrak{E}_k \tag{20}$$

By truncation, it possible to operate on an invariant subspace $\mathfrak{H} = \bigoplus_{k=1}^{q} \mathfrak{E}_k$, thus providing a finite dimensional representation of the operator $\mathcal{T}$ (resp. $\mathcal{K}$), restricted to said subspace, into a matrix $\mathbf{T}$ (resp. $\mathbf{K}$) acting on the vector space $\mathbb{R}^q$. The restriction of a measurement function $\overline{\varphi}^{\mathfrak{H}}$ thus fulfils the linear dynamics

$$\frac{\mathrm{d}}{\mathrm{d}t}\left\{ \overline{\varphi}^{\mathfrak{H}}(\mathbf{x}(t)) \right\} = \mathbf{T}\,\overline{\varphi}^{\mathfrak{H}}(\mathbf{x}(t)) \tag{21}$$

or equivalently

$$\overline{\varphi}^{\mathfrak{H}}(\mathbf{x}(t)) = \exp\{\mathbf{T}\,(t - t_0)\} \cdot \overline{\varphi}^{\mathfrak{H}}(\mathbf{x}(t_0)) = \mathbf{K}^{(t-t_0)}\,\overline{\varphi}^{\mathfrak{H}}(\mathbf{x}(t_0)). \tag{22}$$

## C    Implementation Details

### C.1    State Prediction Tasks

To minimize the alignment loss, the encoder may generate latent codes with arbitrarily small values. We regard these solutions as degenerate, and to discourage this behavior, we normalize the weight columns of the decoder. Moreover, we employ AdamW optimizer by Loshchilov & Hutter (2017) with a slight weight decay of value $1e-4$ and learning rate of $1e-4$. We utilize a custom learning rate of $1e-5$ for the dynamics components, which are either the Koopman matrices or 2-layer MLPs. This encourages the encoder to follow the dynamics prescribed by Koopman and stabilizes training. We use the continuous Koopman parameterization and apply bilinear descretization (Tustin, 1947) whenever control inputs are present. In the absence of control inputs, we allow the model to unroll by solving an initial value problem (IVP), using more sophisticated integrators implemented by JAX (Bradbury et al., 2018). During both training and inference, we make heavy use of the `jax.experimental.ode.odeint()` which relies on an adaptive stepsize (Dormand-Prince) Runge-Kutta integrator (Runge, 1895; Kutta, 1901). We find incorporating the prediction loss to be detrimental to the overall performance when training on the dynamical systems listed in Section 4.1. Inspired by the model described in Appendix D, we use a small sparsity inducing $L_1$ loss, $1e-3$, applied to the Koopman embeddings to encourage region-dedicated dynamics, and use ReLU activations for all experiments to ensure sparse activations. The encoders used for dynamical systems and D4RL are 4-layer and 6-layer standard MLPs, respectively. We use training sequence lengths of 10 and 100, for the dynamical systems and D4RL state prediction tasks, respectively. Moreover, we observe that using reencoding training schemes of 20 and 50 steps are beneficial for most of D4RL state prediction tasks, compared to training without reencoding. Interestingly, presence of control inputs in D4RL dataset makes the training more stable over longer horizons, compared to the dynamical systems we use as benchmark. This is because in the absence of control inputs, the latent state can be arbitrarily scaled up by the eigenvalues of the $\mathbf{K}$ matrix. We find that normalizing the observations and actions as a preprocessing step negatively impacts performance, especially for D4RL tasks. We choose an embedding size of 128 for all dynamical systems, 512 for D4RL states, and 128 for D4RL action embeddings. The $\delta$ stepsize is trained in log space, initialized from that of the environment.

### C.2    D4RL: Semi-Open-Loop Control

In this subsection we review the implementation details that pertain specifically to the open-loop control task. Given that the actions in the `expert` datasets are generated via a fixed expert policy, which is a function of the states, denoted as $u_t = \pi(x_t)$, we can transform the dynamics from their original form, $x_{t+1} = \mathbf{F}(x_t, u_t)$, more compactly into, $x_{t+1} = \mathbf{F}(x_t, \pi(x_t)) = \mathbf{F}_\pi(x_t)$. Therefore, we employ the standard Koopman formulation, i.e. $z_{t+1} = \mathbf{K}z_t$, which does not involve exogenous controls (see Section 3.1). Due to the absence of controls as direct inputs to the model, dictated by the nature of this problem, the training process becomes unstable for long sequences, in comparison to the D4RL state prediction tasks. We train a separate decoder head for action prediction, which takes the encoded states as input. We utilize training sequences with a length of 20 steps, and likewise, employ a periodic reencoding scheme of 20 steps during test time. After the first 100 steps taken by the agent in the environment, we restrict ourselves to observing the ground truth state values only at intervals of 100 steps, hence the designation "Semi-Open-Loop." Additionally, for improved prediction accuracy, we utilize precise adaptive-stepsize integrators implemented in JAX, i.e. `odeint()`, both during the training and inference stages.

## D  SWITCHING DYNAMICS

To explore another limitation of trajectories generated without reencoding, it is instructive to study the special case where $dim(z) > dim(x)$ and $\psi(z) = Wz$, i.e. the decoder is an over-complete dictionary, e.g. a linear *operator*. Therefore the trajectories generated without reencoding correspond to *a linear projection of the solution of a linear dynamical system, e.g.* $x(t) = We^{Kt}\phi(x_0)$. Such trajectories exhibit limited expressiveness; for instance, they cannot capture transitions between multiple distinct fixed points. For example, consider a scenario where $\mathbf{x} \in \mathbb{R}^2$ is governed by a NLDS with multiple fixed points, such as the Duffing Oscillator. Let $\mathbf{z} \in \mathbb{R}^4$. It is possible for $z$ to represent multiple, distinct, attractors of $x$ but not the switching behaviour between them that occurs in the Duffing oscillator for some initial conditions. Let these attractors be located in two regions of state space, $R_1$ and $R_2$. Now, let $\mathbf{z} = [z_1, z_2, z_3, z_4]$ and:

$$\phi(x) = \{[z_1, z_2, 0, 0] \quad \text{if } \mathbf{x} \in R_1, \quad [0, 0, z_3, z_4] \quad \text{if } \mathbf{x} \in R_2\}. \tag{23}$$

These can be interpreted as sparse codes with distinct supports corresponding to $R_1$ and $R_2$, inferred using $\phi$. Furthermore, if we let $K$ be block diagonal then:

$$\dot{z} = \begin{bmatrix} K_1 & 0 \\ 0 & K_2 \end{bmatrix} z. \tag{24}$$

Phase space trajectories are synthesized by linearly combining the trajectories in the latent space.

$$x = \left\{ z_1 \begin{bmatrix} | \\ w_1 \\ | \end{bmatrix} + z_2 \begin{bmatrix} | \\ w_2 \\ | \end{bmatrix} \text{if } x \in R_1, \quad z_3 \begin{bmatrix} | \\ w_3 \\ | \end{bmatrix} + z_4 \begin{bmatrix} | \\ w_4 \\ | \end{bmatrix} \text{if } x \in R_2 \right\} \tag{25}$$

The dynamics of $\mathbf{x}$ can be approximated by two distinct linear systems. Furthermore, the columns of $W$ can shift the origin to represent two distinct fixed points. Note that, without reencoding (Equation 8), the dynamics of $x$ are *restricted to follow the same linear system, determine by the initial condition* $z_0 = \phi(x_0)$, *for all time*. In other words, this scheme does not allow for switching between linear systems, represented by $K_1$ and $K_2$, after $t = 0$. Indeed this is what is observed in Figure 4 (a), the dynamics captured without reencoding resemble the phase diagram of the superposition of several LDS. In contrast, trajectories generated with reencoding, have the capacity to switch between supports (non-zero elements of $z$) and therefore switch between the dynamics defined by $K_1$ and $K_2$. Indeed it was shown that non-linear systems can only be linearized within a basin of attraction of a fixed point (Lan & Mezić, 2013), implying that the best we can hope to achieve is to partition the phase space into regions each of which can be well approximated by LDS, as illustrated in the example above.

## E  EFFICIENCY

Producing full trajectories for $T$ discrete points in a nonlinear dynamical system of the form $x_{t+1} = \mathbf{F}(x_t)$ requires the repeated recurrent application of the nonlinear transformation $\mathbf{F}$. For the Koopman Autoencoder, trajectories for $T$ discrete points are first generated in the Koopman linear space $z_{t+1} = \mathbf{K}z_t + \mathbf{L}v_t$ and followed by a time-agnostic decoding step $x = \psi(z)$. The linear recurrence in the latent space allows for a highly parallelizable unrolling of the predicted sequence $\hat{z}$ using parallel scans (Martin & Cundy, 2018). Compared to nonlinear recurrences, training time is greatly improved. Periodic reencoding does add a nonlinear step to the Koopman autoencoder recurrence. However, since we decode and reencode periodically every $k$ steps, we can parallelize the predicted trajectories for $k$ discrete points and still obtain much improved training efficiencies. Furthermore, considering that our method can provide accurate predictions over short horizons (where reencoding is not needed), it is valuable in situations requiring fast look-ahead capabilities, such as model predictive control (MPC) or $n-$step value function bootstrapping in Reeinforcement Learning.

# F  ADDITIONAL RESULTS

Figure 7 show the error plots for `HalfCheetah-v2` and `Walker2d-v2` environements – `Hopper-v2` error plots can be found in the main text. For all D4RL experiments, during training time, we either do not use reencoding at all, or we use reencoding periods of 20 or 50 steps. The plots correspond to the best performing trained models reported in Table 2. Periodic Reencoding consistently achieves the best results and is robust to the reencoding period.

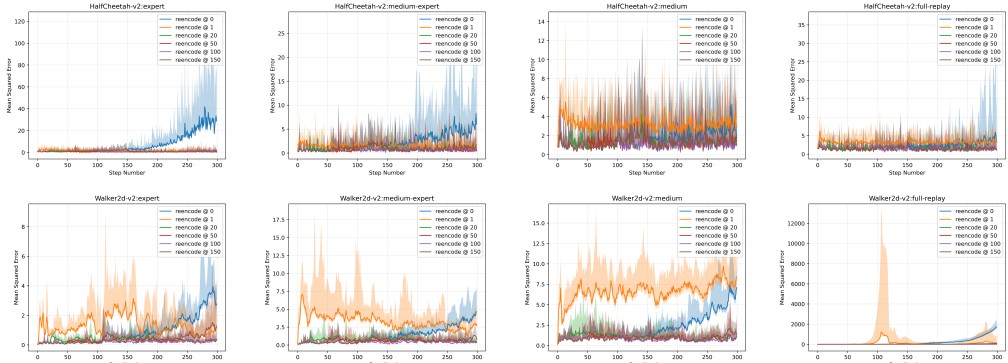

Figure 7: Mean Squared Error (MSE) for HalfCheetah[v2] and Walker2d[v2] environments over the unrolling horizon of 300 steps, under varying reencoding schemes. The results are for Koopman Autoencoder models.

Figure 8 shows the error plots of the Koopman Autoencoder models that DO NOT employ reencoding (periodic reencoding) during training time. We still observe drift in latent space despite the absence of periodic reencoding at training time. Employing periodic reencoding at test time can still mitigate the drift and generate stable, long unrolls.

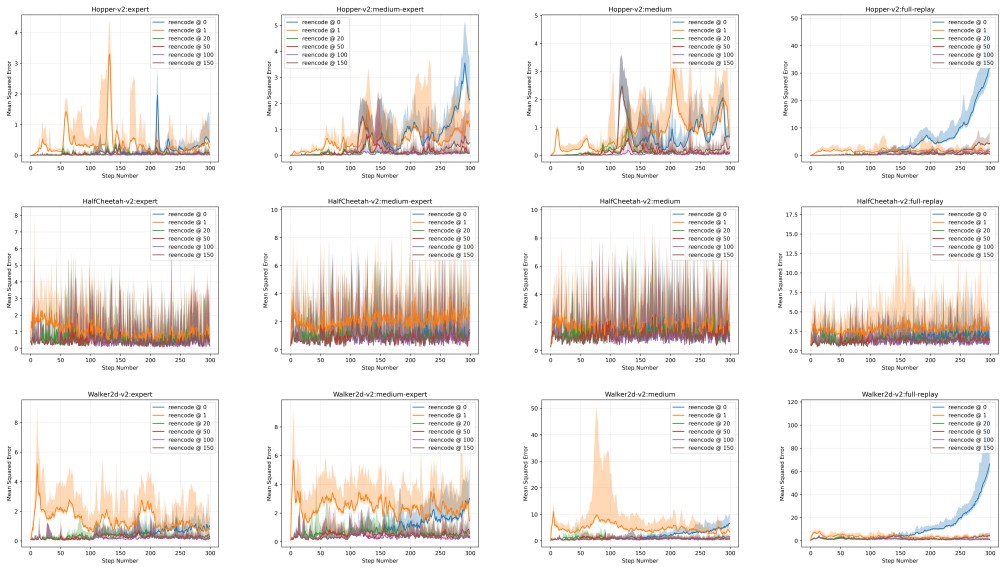

Figure 8: Mean Squared Error (MSE) for D4RL environments over the unrolling horizon of 300 steps, under varying reencoding schemes. The plots belong to models that DO NOT make use of reencoding during training. The errors reported in Table 2, under PERIODIC REENCODING: (✗), corresponds to the blue lines.

**Experiments with Structured Koopman Matrix.** Additionally, we attempted experiments with a diagonal Koopman transition matrix but were unsuccessful. We suspect that this lack of success may be due to additional requirements imposed by the Koopman linear space. Achieving complete disentanglement of features in this context might necessitate an exponentially larger latent state dimension. In an effort to attain training stability and mitigate the issue of scaling up and down the outputs in linear dynamics steps, we conducted experiments using a skew-symmetric matrix, denoted as $K$. Skew-symmetric transition matrices perform rotations without scaling the space. The experiments with the skew-symmetric matrix yielded unsuccessful results, similar to those with the diagonal matrix. We speculate that this lack of success may be attributed to a lack of expressivity.

**Low Dimensional Latents and Deep Latent-to-Latent Mappings.** So far, we have assumed a high-dimensional latent space, where $dim(z) \gg dim(x)$, as in theory, Koopman operator is an infinite-dimensional operator. This enables us to learn rich representations of the dynamics, with the information bottleneck being the linearity requirement by Koopman. We conducted experiments using lower dimensional state embeddings, $dim(x) > dim(z)$, and deep latent-to-latent transition networks. In this context, the information bottleneck more closely resembles that found in tranditional, "static" autoencoders, where the projection into the latent space represents a form of compression. We observed extreme instability when unrolling over long horizons, resulting in poor prediction quality. We also discovered that Periodic Reencoding fails to improve the stability of these models.

**Single Step Koopman Objective.** Rather than training the Koopman objective over multiple steps, we focused on minimizing the losses as defined in Equation 7 at individual timesteps. We implemented this approach with the aim of perfectly aligning the training objective with the inference time scenario, where reencoding occurs at every step. However, this resulted in high inference-time instability and subpar performance, even worse than the MLP baseline, regardless of the utilized inference mechanism, e.g. with or without reencoding. This suggests that the Koopman objective is indeed leveraging its extensibility within basins of attraction, as indicated by Lan & Mezić (2013).

## G  CONNECTION TO RNNS

There exists an extensive body of literature on training RNNs on nonlinear dynamical systems (Pearlmutter, 1990; Trischler & D'Eleuterio, 2016; Vlachas et al., 2022). A prominent application of these models is in Model-Based Reinforcement Learning, where they function as "World Models." For instance, Hafner et al. (2018) introduced Recurrent State Space Models (RSSM) trained on image inputs. These models are often used for short-term future predictions while learning a value function. In contrast, our work facilitates long-range predictions into the future. Koopman autoencoders can be regarded as RNN models with a single linear recurrent layer. From an efficiency perspective, linear recurrent units are more advantageous as they can be executed in parallel rather than sequentially. Furthermore, from a representation learning perspective, one could argue that linearly evolving features extracted from a NLDS would result in a more meaningful and interpretable representation of the system in question. State-space Models (SSMs) (Gu et al., 2022; 2021) can also be regarded as linear RNNs, achieving impressive performance in modeling long-range dependencies. A recent finding by Orvieto et al. (2023) highlights that linear RNN layers surpass tuned nonlinear RNN variants in performance. This result is also partially attributable to the fact that the behavior of linear layers can be designed and engineered more effectively, thanks to the well-studied nature of this problem. This enables effective regulation of linear layers, ensuring the stability of the training process over long sequences. A similar approach was previously adopted by Arjovsky et al. (2015), wherein the eigenvalues of the RNN transition matrix were constrained to lie on the unit circle. The drawback of imposing structure and constraints on the transition matrices is that it results in a loss of expressivity (Monfared et al., 2022).

