# OpenReview forum: "Course Correcting Koopman Representations"
_ICLR.cc/2024/Conference — ICLR 2024 poster_

### Official Review · Reviewer_f4vn · 2023-10-26

**Soundness:** 3 good
**Presentation:** 3 good
**Contribution:** 2 fair
**Rating:** 6
**Confidence:** 2

**Summary:**

Given a fully observable deterministic nonlinear dynamical system described by $dx/dt = f(x)$ such as Lotka-Voltera predator prey population models or chaotic pendulums, Koopman theory states there exists a mapping to a (typically much higher, possibly infinite, dimensional) space $z = g(x)$ where the dynamics are linear, fully described by $dz/dt = Kz$. (I am loosely reminded of the use of kernels in SVMs, a binary classification dataset where classes are not linearly separable can be mapped to a higher dimensional space where classes are linearly separable).

There is an active body of work using the Koopman framework where the mapping function $z = \phi(x)$ is a learnable neural network trained with a corresponding inverse mapping $x = \psi(z)$ which together form an encoder – decoder pair in an autoencoder architecture (albeit where latent $z$ dimension may be lower or higher than the data $x$ dimension), together with this autoencoder, the method learns the transition matrix across time in state latent space $z_{t+1} = K z_t$.

From an initial datapoint $x_0$, time series prediction is performed by encoding the initial point $z_0=\phi(x_0)$, applying the latent state transitions, $z_t = K^tz_0$, and decoding the latent states $x_t = \psi(z_t)$. The authors state that this inference procedure can fail for long term prediction as latent to trajectories that do decode to a sequence of  “drifting” datapoints. Prior works have proposed heuristic solutions such as training on long sequences and restricting the eigen values of the latent space transition matrix K. The authors propose an alternative heuristic they name “periodic re-encoding” where, at every kth transition also incorporates a decoder – encoder pass, i.e.

$z_{t+1} =$

$ K\phi(\psi(z_t))$ if $t \text{mod} k ==0$

$Kz_t \quad \quad \quad$ otherwise

The authors perform experiments on simple well known NLDSs with and without periodic reencoding and make a strong case for the efficacy of periodic re-encoding. The authors apply the method to more challenging datasets from D4RL library and again demonstrate the improvement in performance as a result.


## Update After Rebuttal
I sincerely apologize for missing the response period, I hope the authors can still read this!

- my initial judgement on the value of Koopman representations seems misguided, and the authors and other reviewers responses have significantly increased my appreciation and I have raised my paper score to a weak accept whilst also reducing my confidence score appropriately.

- I now understand the bilinear approach, I did not see the inverse matrix (it as a half step forward and a half step backwards taken in reverse)

**Strengths:**

- the experiments appear to be an ablation study on the effect of periodic re-encoding and show a clear benefit
- the inclusion of both toy and more complex examples is much appreciated.
- there are many works combining autoencoders with latent space transition models. Presumably periodic re-encoding can be applied to any model with the same autoencoder + latent space transition architecture, is it more general than the authors present?

- I felt it was generally well structured and written, although a few minor writing comments mentioned below.

**Weaknesses:**

## Update After Rebuttal
See Summary

# Original Review

For context in the follow up discussion, I would like to state my intuition of Koopman theory is that any finite dimensional nonlinear system can be “lifted” into a higher dimensional space where dynamics are linear, in other words a complex low dimensional nonlinear transition function can be replaced with complex nonlinear high dimensional mapping and a simple transition function, we have just moved the “non-linearity-ness” from one place to another and Koopman theory states that this is universally possible for any NLDS, although in practice this may be very difficult or even impossible in finite dimensions.

## Major Points
- If I understand correctly, when $dim(z) > dim(x)$, the data points encode to a $dim(x)$ manifold embedded in the z-space. One interpretation is that the re-encoding takes latent points that have drifted away from this from the latent data manifold and puts them back on. However, if this were the case, re-encoding every step would have the best results.
- I struggle to agree with the conclusions drawn by the example given in Appendix C.  I view Appendix C as a special case of failure to specify good mappings. The claim that periodic reencoding solves the problem I find strange. If I understand correctly, the perturbations introduced by the decoder-encoder pass allow the latent state to jump between basins of attraction of each fixed point? Presumably these perturbations are just noise and an artefact of the (arbitrary) precision achieved by model training convergence.
- what is the benefit of periodic re-encoding, clamping latent representations to the data manifold or adding noisy perturbations?
- the authors include non-linear transitions as a baseline, while it is nice to have more baselines, as I understand, linear transitions is the only reason to use Koopman theory. As expected, the (strictly higher capacity) non-linear transition outperform the liner transitions.
- Given the re-encoding is not specific to Koopman, and that Koopman (linear) transitions predictably perform worse than non-linear transitions, this paper’s focus on the Koopman framework seems unnecessary.
- there are many many autoencoder + latent space transition models that vary mainly by application specific architectural differences (e.g. pixels/graph/vectors require CNN/GCN/MLP encoder) and their training losses (L2, ELBO, Cross entropy etc), and a quick google for "latent dynamics" and "deep kalman" yields many results, many that have linear latent transitions
  - https://papers.nips.cc/paper_files/paper/2017/hash/7b7a53e239400a13bd6be6c91c4f6c4e-Abstract.html
  - http://www.approximateinference.org/2015/accepted/KrishnanEtAl2015.pdf
  - https://arxiv.org/abs/1811.04551



## Minor Points
- Section 2, first paragraph does not help unfamiliar people understand Koopman theory or MDM (and familiar people do not need to be told). I would suggest dumbing this down as the level of Koopman theory required for this paper can be explained very easily.
- Bibliography: many of the references have titles, names and dates but are missing journal/conference information
- Appendix E is a nice addition, yet it feels somewhat arbitrary to cite a handful or RNN papers when the time series modelling literature is vast and the autoencoder + latent dynamics literature is I feel is more closely related this work than RNNs.

**Questions:**

- For each problem, can you state the dimensions of $x$ and $z$ in the main paper. (I couldn’t see them in Appendix B)
- given linear latent transitions are worse and periodic reencoding is more general than Koopman autoencoders, what is the justification for the authors to focus on Koopman representations? What is the benefit of the Koopman framework over any other autoencoder based time series model?
- Equation 6: it appears the transition matrix is applying ½ of a forward time step and then ½ of a backward time step? I would understand if the transition matrix was made of many incremental time steps e.g. $\textbf{K}=(I + \frac{\delta}{n}K)^n$ where $n =2$ (and consequently $\textbf{K}=exp(\delta K)$ as $n>\infty$)
- what is the benefit of periodic re-encoding, clamping (high dim) latent representations to the latent data manifold or adding noisy perturbations?

---

> ### Author Response · Authors · 2023-11-16
> **Response to R-f4vn [1 / 2]**
>
> Thank you for your thorough and careful review of our work. Your summary captures the essence of our proposed method, and we appreciate your recognition of its strengths. In this response, we have addressed the major points and will proceed to address the minor points and questions in the following response.
>
> ---
>
> > “...linear transitions is the only reason to use Koopman theory. As expected, the (strictly higher capacity) non-linear transition outperform the liner transitions.”
>
> > “...this paper’s focus on the Koopman framework seems unnecessary.”
>
> The primary motivation for studying Koopman autoencoders is their ability to ***learn rich, interpretable representations of nonlinear dynamical systems, a capability anchored in the linearity requirement of the Koopman theory***. Indeed, our paper presents several key results pertaining to this setting. Although our paper used periodic reencoding to overcome limitations related to linear transition dynamics (**Equations 8 & 9**, **Appendix C**) – it is nevertheless applicable to more general latent dynamics models that share some of the same limitations.
> Although we cannot use the closed form solution to linear dynamical systems to derive Equation 8, Equation 9 still holds for nonlinear latent dynamical models ($\mathit{K}$ would just be a nonlinear function of $x$). These models suffer from the same limitations as Koopman autoencoders -- they also violate the IVP without reencoding.
>
> We agree with the reviewer that models with nonlinear latent state dynamics are indeed
> less meaningful/interpretable than the linear latent dynamics, nevertheless we feel these experiments add value to the paper by broadening the applicability of our proposed method.
>
> Furthermore, there has been an increasing interest in the study of linear recurrent models recently, primarily due to their efficiency and the ability to run them in parallel using convolution. These models have demonstrated impressive results in tasks requiring long-range reasoning, as evidenced by the performance of State-Space Models. Notably, Koopman Autoencoders bear a close relation to linear recurrent models, a connection further explored in **Section E.1.2** of [arxiv.org/abs/2303.06349](https://arxiv.org/abs/2303.06349). A key contribution of our work is to scale Koopman Autoencoder to more challenging problems, by addressing the limitations pertaining to their particular setting.
>
> ---
>
> > “If I understand correctly, the perturbations introduced by the decoder-encoder pass allow the latent state to jump between basins of attraction of each fixed point?”
>
> Yes, this is the idea presented in **Appendix C** and we discuss this in the "[General Response](https://openreview.net/forum?id=A18gWgc5mi&noteId=ddMyUvKB7n)." However, we disagree with the use of the term “perturbation” which implies a random change of support. The sparse Koopman Autoencoder model presented in **Appendix C** implicitly learns to assign a distinct support for each region of state space. Supports are not “randomly” assigned, but must correspond to the appropriate linearized representations corresponding to each region of the state space.
>
> ---
>
> > “what is the benefit of periodic re-encoding, clamping latent representations to the data manifold or adding noisy perturbations?”
>
> Again, we disagree with the use of the term “noisy perturbation," please see our comment above. Periodic reencoding solves both issues: **1.** it ensures that the IVP principle is satisfied by construction, and **2.** it enables switching between representations corresponding to basins of attraction, i.e. proved in Lan & Mezic.
>
> ---
>
> > “One interpretation is that the re-encoding takes latent points that have drifted away from this from the latent data manifold and puts them back on.”
>
> Yes, this is a good way to describe issue (**2.**) discussed above. Indeed the results in **Figure 4(a)** (top left) serve as a confirmation of this. The results resemble the superposition of the phase plots of multiple linear dynamical systems, because this is exactly what is produced by linear latent dynamics and linear decoder without reencoding. The solutions are invalid over long horizons because of _overexploitation using a stale linearized representation_. Reencoding infers a new support and thereby updates the linearized representation.
>
> ---
>
> > “However, if this were the case, re-encoding every step would have the best results.”
>
> Yes, we agree that this reprojection onto the data manifold is a good interpretation. However the reprojection is not perfect, and _introduces its own error_. Rolling out the dynamics in a regularized latent space for limited horizons is one way to mitigate the accumulation of such errors. In a way, reencoding, while not perfect, is a tool we use in moderation to address the issue of drift. This approach aligns with the findings of Mezic et al., who demonstrated that the Koopman operator can be locally extended within basins of attraction.

---

> ### Author Response · Authors · 2023-11-16
> **Response to R-f4vn [2 / 2]**
>
> In this response we address your minor concerns and answer the questions you had.
>
> ---
>
> > Existing Latent Space Transition Models
>
> The aim of the paper is to establish the issue of drift of latent states in Koopman Autoencoders and to present periodic reencoding as a simple and effective solution. We believe the effectiveness of periodic reencoding further sheds light on the implications of latent space trajectory generation, while readily being applicable to similar settings.
>
> ---
>
> > “What is the benefit of the Koopman framework over any other autoencoder based time series model?”
>
> We hypothesize that simpler latent transition mappings and decoder structures lead to **richer representations** of the dynamical system in question. To make accurate future predictions, the encoder needs to encapsulate and disentangle future state information into the current latent state as much as possible. Koopman theory shows that such linear mappings can indeed well approximate nonlinear systems, provided that they are of high dimensionality, ideally infinite-dimensional. The prescribed linear dynamical system is not only _easier to study_ but also offers greater **interpretability**. Additionally, it enables the **application of linear tools**, such as Linear Quadratic Regulator (LQR), for control purposes. Furthermore, linear dynamical systems, whether continuous or discrete, can be integrated/unrolled in time with **high efficiency and in parallel**. These factors collectively serve as strong motivators for conducting research on such systems.
>
> ---
>
> > “Section 2, first paragraph does not help unfamiliar people understand Koopman theory or MDM (and familiar people do not need to be told). I would suggest dumbing this down as the level of Koopman theory required for this paper can be explained very easily.”
>
> We agree. Thanks for your constructive feedback. We are posting the responses now to allow enough time for discussion regarding the main aspects of the paper. For the camera-ready version, we acknowledge that Section 2 needs a revision. We plan to undertake this revision, keeping in mind that it requires time.
>
> ---
>
> > “the autoencoder + latent dynamics literature is I feel is more closely related this work than RNNs.”
>
> Latent dynamics models are indeed closely related to recurrent neural networks, with the distinction between the two often being subtle. For instance, the _“world model”_ referenced in the work you cited ([arxiv.org/abs/1811.04551](https://arxiv.org/abs/1811.04551)), which is a latent dynamics model, is termed as “Recurrent State Space Model.” In light of this, we agree with your observation and **have updated our manuscript** to include a discussion on the relationship between these two approaches in the same Appendix section.
>
> ---
>
> > “Bibliography: many of the references have titles, names and dates but are missing journal/conference information”
>
> We have fixed the bibliography to include the publication details.
>
> ---
>
> > “For each problem, can you state the dimensions of and in the main paper. (I couldn’t see them in Appendix B)”
>
> Information regarding latent space dimensions can be found in the concluding sentences of **Appendix B.1**. Additionally, we have updated the manuscript to include this information in the main text for easier accessibility.
>
> ---
>
> > “Equation 6: it appears the transition matrix is applying ½ of a forward time step and then ½ of a backward time step? I would understand if the transition matrix was made of many incremental time steps e.g. $\mathbf{K} = (I + \frac{\delta}{n} K)^n$ where $n = 2$ (and consequently $K=exp(\delta K)$ as $n > \infty$”
>
> This is a standard discretization method used widely called blinear discretization, which S4 also uses.
>
> A simple derivation would be: $\quad e^{K \delta} = e^{K \delta /2} (e^{-K \delta /2})^{-1} \approx (I + K \delta /2)(I - K \delta /2)^{-1}.$
>
> This is an instance of a standard method in the field of numerical resolution of ordinary differential equations (ODEs): the implicit midpoint rule. It is well documented in the literature, one reference being ["Solving Differential Equations 1" by Sylvert Paul Nørsett, Gerhard Wanner & Ernst Hairer, p204 Equation 7.4 & p205 Table 7.1]
>
> ---
>
> We hope our responses have clarified your concerns. If you are satisfied with our updates, we would appreciate your consideration in revising the score. Thank you for your valuable feedback and support.

---

> > ### Author Response · Authors · 2023-11-23
> > **Awaiting Response from R-f4vn**
> >
> > _The official public discussion phase is ending today._
> >
> > We kindly ask you to consider our general and specific responses.
> >
> > We believe that we have addressed all of your questions.
> >
> > Thank you.

---

### Official Review · Reviewer_Woag · 2023-10-30

**Soundness:** 4 excellent
**Presentation:** 4 excellent
**Contribution:** 4 excellent
**Rating:** 8
**Confidence:** 3

**Summary:**

This work we study Koopman autoencoders to model nonlinear dynamics, specifically for long term state forecasting. The authors propose Periodic Reencoding, an inference-time mechanism that incorporates far future (multiple-step ahead) loss for faithfully capturing long term dynamics. The proposed method is validated both analytically and empirically via experiments in low and high dimensional NLDS.

**Strengths:**

The proposed Periodic Reencoding improves long term forecasting of deep Koopman autoencoders.

It does not require long sequence training.

The manuscript is well written.

**Weaknesses:**

The Periodic Reencoding works well especially with oscillator dynamics. It is likely related to the fact that those dynamics is periodic. It would be interesting to show how it works with continuous attractors.

**Questions:**

It's worthy discussing the "optimal" $\Delta t$ and its relationship to the dynamics. For example, Fig. 5 shows the best are the middle ones.

---

> ### Author Response · Authors · 2023-11-16
> **Response to R-Woag**
>
> Thank you for your review! We have answered to your questions in the follwoing response.
>
> ---
>
> > “The periodic reencoding works well especially with oscillator dynamics. It is likely related to the fact that those dynamics is periodic. It would be interesting to show how it works with continuous attractors.”
>
> We show that our method can successfully capture the dynamics of a system characterized by chaotic attractors, such as Lorenz’63 and walking/running robots (D4RL). We think that an adaptive mechanism that adjusts the reencoding period is worth investigating for such systems.
>
> ---
>
> > “It's worthy discussing the "optimal" $\Delta t$ and its relationship to the dynamics. For example, Fig. 5 shows the best are the middle ones.”
>
> Mezic et al. demonstrated that the Koopman operator can be locally extended within the basins of attraction of fixed points. As detailed in **Appendix C** of our paper, reencoding can function as a switch between the linear Koopman dynamics governing each region. Therefore, in theory, reencoding becomes necessary when trajectories traverse the boundaries between these basins of attraction. This suggests that the optimal reencoding period needs to be adjusted according to the rate at which we jump from one region to the other.
>
> A fascinating future research would be to develop a method to learn the optimal timing for reencoding, rather than relying on a fixed, periodic schedule. This approach could lead to a more adaptive mechanism. Toward this goal, we have been experimenting with training a reencoding policy network through Reinforcement Learning, which is still a work in progress. This network, conditioned on the latent state, outputs the decision of whether to reencode at each step. However, as of now, periodic reencoding still demonstrates superior performance compared to our implementation of this method.

---

> > ### Comment · Reviewer_Woag · 2023-11-20
> >
> > Thank the authors for the response. My concerns have been addressed.

---

### Official Review · Reviewer_Fg8g · 2023-11-03

**Soundness:** 3 good
**Presentation:** 3 good
**Contribution:** 2 fair
**Rating:** 6
**Confidence:** 3

**Summary:**

This paper introduces the concept of periodic reencoding into the field of Koopman autoencoders. The idea is that, rather than propagating the dynamical system solely in the latent space, we periodically re-encode the true-space state $x_t$ to the latent space. This overcomes possible issues with $\phi$ and $\psi$ not being exact inverses (due to lack of bijectivity).

**Strengths:**

- the concept of periodic re-encoding is clear and simple
- it leads to across-the-board improvements in performance

**Weaknesses:**

- any underlying mechanism governing periodic re-encoding is not discussed. For example, are there deficiencies in the encoder or decoder that, if adjusted, remove the need for reencoding, or change the optimal period?
- how is the optimum period chosen? Since every step reencoding is worse in terms of performance than periodic reencoding, this is not a simple tradeoff in terms off costs
- what causes do we hypothesise for "long term drift"? Perhaps during training the encoder does not learn to "use" all the latent space. Thus, there exist points in latent space with a decoding that is not properly trained. This is pure conjecture, but it would be beneficial for this paper to investigate these issues.
- text is not legible in some figures (Fig 3, 4, 5 font size too small, Figure 6 more extreme)

**Questions:**

- has equation (2) been used in previous papers? I see that Lusch et al. (2018) used something similar. It would be helpful to precisely clarify.

---

> ### Author Response · Authors · 2023-11-16
> **Response to R-Fg8g**
>
> Thank you for your careful review of our paper. We believe that the following response addresses your concerns and clarifies key aspects of our work. We also encourage you to review our "[General Responses](https://openreview.net/forum?id=A18gWgc5mi&noteId=ddMyUvKB7n)" and other responses to `R-QVMC` and `R-FYpR`, as we find them highly relevant to your questions as well.
>
> ---
>
> > “any underlying mechanism governing periodic re-encoding is not discussed. For example, are there deficiencies in the encoder or decoder that, if adjusted, remove the need for reencoding, or change the optimal period?”
>
> > “what causes do we hypothesise for "long term drift"?”
>
> One of the key issues with trajectory generation in latent space without reencoding stems from the lack of injectivity of the mappings from $z$-space into $x$-space. This lack of injectivity arises directly from the Koopman assumption that, $dim(z) > dim(x)$, a fundamental aspect of the setting. When reencoding is not employed, _trajectories_ in $z$-space are entirely decoded to _curves_ to $x$-space. Due to the non-injective nature of the mappings into $x$-space, the curves will cross over themselve, and this violates the IVP principle. This phenomenon occurs because multiple points along the $z$-trajectory are mapped to the same point in $x$ space, i.e. the crossover point. A simple diagram is [linked here](https://i.imgur.com/c7Hrv8d.png).
>
> With reencoding, we are modeling the NLDS with a learned feedback (in $x$) which generates trajectories that do not cross by construction. The existence of the crossover point contradicts the fundamental principles of dynamical systems, suggesting that this method of inference is unsuitable. This deficiency, together with the inability to capture multi-attractor dynamics (**Appendix C**) is the source of what we refer to as 'long term drift' in the model's predictions. ***These issues cannot be overcome by architectural modifications to feedforward encoders/decoders.***
>
>
> We establish that reencoding gives rise to valid solutions that do not violate the IVP principle; however, when used at every step, reencoding errors will compound and not follow the true trajectory of the system. The question of exactly why reencoding should be done only occasionally warrants further study, but we hypothesize that the reprojection is imperfect and can also accumulate errors. Thus we introduce periodic reencoding as a middle ground between no reencoding and reencoding at every step. We show empirically that this process generates the best predictions over long horizons.
>
> ---
>
> > “how is the optimum period chosen? Since every step reencoding is worse in terms of performance than periodic reencoding, this is not a simple tradeoff in terms off costs”
>
> Our experiments empirically show that there is an optimal reencoding period, which should be tuned for every NLDS if we set aside computational cost and focus solely on prediction quality. We show that the method is relatively robust to the reencoding period as well (see error plots under varying reencoding schemes in the paper). One important aspect of our work is that it offers a spectrum from which the optimal reencoding period can be selected, according to the available computational budget and desired prediction quality. The computational budget may also vary based on the specific encoder/decoder used. The entries in the tables are based on the best reencoding period in terms of accuracy.
>
> ---
>
> > “text is not legible in some figures”
>
> Thank you for bringing this to our attention. We will increase the font size for the camera-ready version.
>
> ---
>
> > “has equation (2) been used in previous papers? I see that Lusch et al. (2018) used something similar. It would be helpful to precisely clarify.”
>
> We have updated the manuscript to mention previous works optimizing the same objective, i.e. Equation 2.
>
> ---
>
> _We trust that we have addressed most of your concerns and hope this may lead you to reconsider your assessment. We are open to and welcome any further discussions._

---

> > ### Author Response · Authors · 2023-11-23
> > **Awaiting Response from R-Fg8g**
> >
> > _The official public discussion phase is ending today._
> >
> > We kindly ask you to consider our general and specific responses.
> >
> > We believe that we have addressed all of your questions.
> >
> > Thank you.

---

> > > ### Comment · Reviewer_Fg8g · 2023-11-23
> > > **Reply to authors' response**
> > >
> > > Thank you very much for your response.
> > >
> > > > The question of exactly why reencoding should be done only occasionally warrants further study
> > >
> > > I agree with this sentiment, which was at the heart of my concern. It would be great to be able to understand this more definitely.
> > >
> > > > These issues cannot be overcome by architectural modifications to feedforward encoders/decoders
> > >
> > > Is this true even when the encoders/decoders are non-deterministic? In such a case, the strict arguments about injectivity would no longer apply.

---

### Official Review · Reviewer_FYpR · 2023-11-04

**Soundness:** 3 good
**Presentation:** 3 good
**Contribution:** 3 good
**Rating:** 6
**Confidence:** 2

**Summary:**

This work proposes an autoencoder model for dynamical system data based on Koopman theory; observed variables are mapped to the "Koopman feature" space in which the state evolution is linear.  The authors proposed a periodic re-encoding scheme that avoid the issues of trajectory crossing in observation space and the inability to model switching dynamics.  The authors demonstrated that the proposed method with re-encoding leads to improved state prediction and controlling.

**Strengths:**

- This paper is mostly well-written.
- The proposed method demonstrates promising empirical performance.

**Weaknesses:**

- I feel the theoretical benefits of re-encoding are not well explained.  It appears that re-encoding should be most useful in eliminating undesirable local optimas; if the model is correctly specified, the AE objective should have the same global optima with or without re-encoding.  It is thus not clear whether it is a principled solution to the switching dynamics issue, which is one of the two motivations for this work; if the dynamics cannot be (globally) linearized the model is misspecified in either case.

- While the experiments demonstrate the benefits of re-encoding for autoencoding approaches, it is not clear why we should restrict to such approaches for control. There is also a general lack of comparison with baselines, e.g. with Mondal et al (2023).

- As a minor point, the reference to deep state-space models such as S4 is slightly misleading as they are often applicable to stochastic systems, whereas this work studies deterministic systems.

**Questions:**

See above.

## Post-rebuttal update

Thank you for your response.  I am raising the score as my concern is largely addressed.  I remain uncertain about the interpretation of the RL experiments; while they have indeed demonstrated the utility of the proposed method in the Koopman framework, the lack of baseline comparisons makes it somewhat difficult to put them in context, i.e. to what extent it is addressing an issue in an application where the Koopman framework is of interest.  As I am not very familiar with this area, I will defer the judgement of this issue to the other reviewers.

---

> ### Author Response · Authors · 2023-11-16
> **Response to R-FYpR [1 / 2]**
>
> We appreciate your insightful feedback and careful review of our work. Your comments have been thoughtfully incorporated into the manuscript, and we believe they have enhanced our presentation of both the method and the problem setting.
>
> ---
>
> > “It appears that re-encoding should be most useful in eliminating undesirable local optimas; if the model is correctly specified, the AE objective should have the same global optima with or without re-encoding.”
>
>
> The main challenge that we aim to address via periodic reencoding, is to ***unroll the trained model over extended horizons***, i.e. longer than the sequence lengths that can be used at training time. If the objective was to predict future states for the same number of steps as in the training sequence length, then the problem is “correctly specified” and use of periodic reencoding would be unnecessary. This also necessitates the assumption that the model is equipped with a _nonlinear_ decoder with large capacity, which prevents the system from learning rich latent representations. In our work we analytically show that periodic reencoding is absolutely necessary, _even for predictions within the training sequence length_, when a linear decoder is used (**Section 3.2** and **Appendix C**).
>
>
> Periodic reencoding is crucial for accurately predicting future states ___beyond the scope of the training sequence___, even in the general setting with assumed nonlinear components. This necessity is substantiated both empirically and theoretically, as detailed in **Section 3.2** of our manuscript, where we show $x$-space trajectories, generated without reencoding, violate the Initial Value Problem setting. We kindly refer you to our post "[General Response](https://openreview.net/forum?id=A18gWgc5mi&noteId=ddMyUvKB7n)" for a more detailed explanation.
>
>
> As an alternative, one can choose to train the model with _nonlinear_ components over long sequences; however, this imposes hardware requirements and requires additional tricks to stabilize gradients. A number of papers cited in our work achieve this by engineering the latent dynamics. For example Unitary RNNs force the eigenvalues of the latent-to-latent transition to lie on the unit circle. These constraints reduce the expressivity of the model (arxiv.org/abs/2110.07238). We do not impose such constraints on the latent transition matrix.
>
>
> For all dynamical systems, we find training over horizons longer than 10 steps (and 100 steps for D4RL tasks) to be extremely unstable unless the parameters of $K$ are somehow constrained. However, through periodic reencoding we can unroll the model **up to 1000 steps stably and accurately**. We use a small training dataset of 50 trajectories of length 500 steps and use training sequences of length 10, however, we are still able to unroll the model successfully for 1000 steps – **500 steps longer than in training**.
>
> ---
>
> > “if the dynamics cannot be (globally) linearized the model is misspecified in either case.”
>
>
> According to Koopman theory, the operator $K$ drives the Koopman embedding of the states forward _infinitesimally_ in time, given a freshly encoded point. In other words, the Koopman matrix is **spatially global**, as it is fixed/constant and applicable to all encoded points, however **temporally infinitesimal**. _Lan and Mezic show that the **Koopman operator can be applied locally rather than infinitesimally**, as it can be extended to the basin of attraction corresponding to the initially encoded point_. **Appendix C** illustrates how this idea can be implemented in practice, as sparse Koopman autoencoder. We use periodic reencoding to mitigate this issue, which in your review is refered to as "misspecification," all the while enjoying the benefits provided by Koopman -- rich representations, efficiency, interpretability, etc.

---

> ### Author Response · Authors · 2023-11-16
> **Response to R-FYpR [2 / 2]**
>
> > “it is not clear why we should restrict to such approaches for control.”
>
>
> In our work we aim to highlight the limitations of the Koopman framework and introduce periodic reencoding as a way to counter them. We do not claim that periodic reencoding is a replacement to other innovations, nonetheless our model/approach attains high prediction quality. We use D4RL as an standard benchmark that is widely used to allow such comparisons at the discretion of the reader. The sole motivation for conducting the semi-open-loop control experiment is to provide readers with a more tangible metric. In contrast to state mean squared error (MSE), we offer the normalized reward achieved by the agent as a more relevant measure to downstream applications (e.g. semi-open loop control).
>
> Furthermore, Mondal et al. (can be considered as concurrent work according to the guidelines) report future state prediction error over the horizon of 100 steps for D4RL tasks, which is also the sequence length used during training. The challenge that our method addresses is to ***go beyond the training sequence at inference time***, hence we report the error for **300 steps** – which is **still better/comparable to 100-step error of Mondal et al**.
>
> ---
>
> “...the reference to deep state-space models such as S4 is slightly misleading as they are often applicable to stochastic systems, whereas this work studies deterministic systems”
>
> We have updated our manuscript to ensure that the comparisons to State-Space Models only pertain to efficiency gains from linear recurrence. That being said, State-Space Models have been applied to deterministic systems, NLDS, D4RL and similar control environments (see [arxiv.org/abs/2306.05167](https://arxiv.org/abs/2306.05167)).
>
> ---
>
> _We trust that we have addressed most of your concerns and hope this may lead you to reconsider your assessment. We are open to and welcome any further discussions._

---

> > ### Author Response · Authors · 2023-11-23
> > **Awaiting Response from R-FYpR**
> >
> > _The official public discussion phase is ending today._
> >
> > We kindly ask you to consider our general and specific responses.
> >
> > We believe that we have addressed all of your questions.
> >
> > Thank you.

---

### Official Review · Reviewer_QVMC · 2023-11-05

**Soundness:** 4 excellent
**Presentation:** 3 good
**Contribution:** 3 good
**Rating:** 8
**Confidence:** 4

**Summary:**

The authors of the study investigated the use of Koopman autoencoders to model complex, nonlinear dynamical systems. They explored different approaches for making long-term predictions and showed that generating predictions solely within the latent space had two potential challenges: 1) It was unable to effectively capture the behavior of multi-stable systems that switch between multiple fixed points. 2) It violated the existence and uniqueness theorem of initial value problems, which establishes the uniqueness of solutions in dynamical systems. To address these challenges, the authors introduced a technique called "periodic reencoding." This method strikes a balance between not using reencoding at all and reencoding at every step, and it was found to yield the best practical results in their experiments.

**Strengths:**

- The paper is well-written, featuring a fascinating main idea, and employing innovative and effective methods with good results.

- The overview provided in the appendix is very helpful and is written in a precise and technical mathematical style.

- The visualization provided in Fig. 1 is very helpful in providing a clear illustration of their method.

**Weaknesses:**

**A minor point**:  Regarding the sentence "This characteristic becomes especially significant when the dimensionality of the latent space is substantially larger than that of the original space, i.e., n >> d, ...", why is this case considered more important? While it's true that a mapping to a larger dimensional space is not surjective, as discussed in the paper, it seems that the lack of injectivity of the map is more critical. So, since a mapping to a smaller dimensional space is not injective, wouldn't the case of n << d be more problematic?

**A few questions**:

1. When it comes to learning chaotic dynamics, what are the advancements offered by the proposed method compared to using RNNs with teacher forcing, as presented in the following papers?

[1] F. Hess, Z. Monfared, M. Brenner, D. Durstewitz, Generalized Teacher Forcing for Learning Chaotic
Dynamics, Proceedings of Machine Learning Research (ICML 2023), (2023).

[2] J. Mikhaeil, Z. Monfared, D. Durstewitz, On the difficulty of learning chaotic dynamics with RNNs, Advances in Neural Information Processing Systems (NeurIPS), (2022).

2. Regarding eq. (9), it apears that the Jacobian $J_{\psi}$  of  $\psi$ is assumed to be a constant matrix. If this is the case, how can one ensure that it remains a fixed matrix? It may vary depending on the specific points at which it is calculated and could differ across different points.

3. How effectively would the proposed method perform in capturing the dynamics of a multi-stable system with a stable fixed point and a periodic attractor, or possibly with two periodic attractors?

4. How changes in the hyperparameters $\Delta t$ or $k$ can impact the results? Does an optimal value always exist for that?

5. Why do the authors use an MLP instead of K (e.g., on page 6)? How does the choice of activation function affect the performance of the MLP? Were other activation functions considered, and if so, how did they compare to the chosen function? Can the MLP be replaced with other types of neural networks, such as convolutional neural networks or recurrent neural networks? How would this affect the performance of the proposed framework?

I am happy to increase my score if the authors could address my concerns.

**Questions:**

Please see above!

---

> ### Author Response · Authors · 2023-11-16
> **Response to R-QVMC [1 / 2]**
>
> We appreciate the recognition of our work’s strengths, the on-point summary and your insightful feedback. We concur that the simplicity and practicality of periodic reencoding facilitates its use in any approach that involves latent space dynamics. By correcting for drift, periodic reencoding more faithfully captures long term dynamics compared to never/every-step reencoding approaches. If the latent dynamics are assumed to be linear between reencoding steps, it also enables computationally efficient unrolling of dynamics.
>
> In **Section 3.2**, we highlight that models with linear dynamics necessitate reencoding intervention for faithful capturing of the dynamics over long horizons (Equations 8-9). **Appendix C** demonstrates that Koopman dynamics are inadequate for capturing trajectories that extend beyond their originating basin of attraction. It proposes periodic reencoding as an elegant solution to this limitation. For a more details kindly refer to our "[General Response](https://openreview.net/forum?id=A18gWgc5mi&noteId=ddMyUvKB7n)."
>
> ---
>
> > “why is this (n >> d) case considered more important?”
>
>
> In theory, the Koopman operator is an infinite-dimensional mapping between latent codes. However, for practicality, Koopman Autoencoders are learned end-to-end by assuming a high-dimensional but finite latent space, i.e., $dim(z) \gg dim(x)$ or $n \gg d$.
>
> Trajectories generated in the latent space are valid, in the sense that they adhere to the initial value problem (IVP) and do not cross. This stems directly from the equation $z_{t+1} = K z_t$, which defines a (linear) dynamical system. However, when reencoding is not utilized during inference, and trajectories from the latent space are simply decoded to produce _curves_ in the original space, a discrepancy arises. Unlike the latent space trajectories, these curves can cross, thereby _violating the IVP property, at the crossover point_.
>
> Due to the non-injective nature of the mapping from the latent space to the original space, _multiple_ points in the latent space can map to the _same $x$_, the crossover point, in the original space. This property is _unavoidable_ when $n > d$, i.e. mappings from higher dimensional to lower dimensional space cannot be injective. Nevertheless, even if $d > n$, the mapping is not necessarily injective and we must rely on statistical learning to capture this constraint.
>
>
> ---
>
> > “...proposed method compared to using RNNs with teacher forcing”
>
>
> Thank you for the references; they are indeed relevant. Please see our post “[General Response](https://openreview.net/forum?id=A18gWgc5mi&noteId=ddMyUvKB7n)”. In summary, periodic reencoding can be used _in conjunction_ with many existing methods and _and should not be viewed as a competing method_. ***Periodic reencoding addresses fundamental limitations of modelling feedback processes with feed-forward functions -- this includes not satisfying the IVP property.***
>
>
> From a high-level perspective, periodic reencoding can be used as an inference-time method for models with a single recurrent layer, i.e. the latent state dynamics, while [1] (concurrent and new to us) and [2] (discussed in **Appendix F**) studies are more concerned with the gradient behavior during training time applicable to all RNNs.  However, similar to [2], we also do use a “sparse” teacher-forcing regime during training time, as we optimize the loss over multiple future steps, given only the ground truth initial condition as input (see Figure 1). We reiterate that periodic reencoding is conceptually different from teacher forcing, as stated in the paper. Furthermore, [1] rectifies gradients by interpolating between the teacher forced and model-generated latent states. On the other hand, periodic reencoding, when employed at inference time, addresses a fundamental limitation of Koopman autoencoders, in regards to Koopman autoencoders. We show that our method can successfully model chaotic dynamics, Lorenz’63 and D4RL environments, all the while achieving this using a single linear recurrent layer, $\mathit{K}$.

---

> ### Author Response · Authors · 2023-11-16
> **Response to R-QVMC [2 / 2]**
>
> > “...the Jacobian is assumed to be a constant matrix”
>
>
> The Jacobian varies as a function of $x$. We do not assume a constant Jacobian and we have updated our notation to reflect this properly. The only assumption that we draw conclusions from is that the RHS of Equation 9 is a function of $x$, forming an instant feedback loop in dynamics.
>
> ---
>
> > “dynamics of a multi-stable system”
>
> Please see our post “[General Response](https://openreview.net/forum?id=A18gWgc5mi&noteId=ddMyUvKB7n).” _**Appendix C** illustrates exactly such an example_ and demonstrates that reencoding is absolutely necessary to capture multi-attractor dynamics when using a linear decoder. This is more difficult to prove for a general decoder, but nevertheless the example conveys the intuition of what makes this problem difficult. Furthermore, our model demonstrates high-quality prediction performance on the low-/high-dimensional, controlled, and chaotic dynamical systems, covering a broad spectrum of dynamics.
>
> ---
>
> > “changes in the hyperparameters $\Delta t$ and $k$”
>
>
> In our experiments, longer training horizons, $k$, resulted in better performance, whereas shorter training sequences led to poorer outcomes. Therefore, we chose $k$ solely based on the stability of training and hardware constraints. $\Delta t$ is the duration between two reencoding operations at inference time. We show that there exists an optimal $\Delta t$, given that never/every-step reencoding schemes result in poor prediction quality.
>
> ---
>
> > “Why do the authors use an MLP instead of K?”
>
> To clarify, we indeed conduct experiments using the K matrix for latent-to-latent mapping, and the results of these experiments are reported in all tables under the 'Koopman Autoencoder' section. The entries labeled 'Nonlinear Latent Dynamics' in **Table 1** pertain to the models equipped with a _wide MLP_, as the latent-to-latent mapping. Although periodic reencoding was initially motivated for models with linear latent transitions dynamics, our empirical findings demonstrate that periodic reencoding is also useful when using non-linear dynamics, which we believe further broadens the applicability of our work to more general models.
>
> Regarding activation functions, in our experiments with low-dimensional dynamical systems, ReLU achieved slightly better results than ELU. We attribute this to induced sparsity by ReLU, by the intuition from the example in **Appendix C**. We have not conducted experiments with various architectures for latent-to-latent mappings, but we have reasons to believe that periodic Reencoding would still enhance the prediction quality.
>
> ---
>
> _We trust that we have addressed most of your concerns and hope this may lead you to reconsider your assessment. We are open to and welcome any further discussions._

---

> > ### Comment · Reviewer_QVMC · 2023-11-21
> > **Thanks!**
> >
> > I appreciate the authors for their response and clarification! After reading other reviews and their responses, I am inclined to stick to my original score.

---

> > > ### Author Response · Authors · 2023-11-21
> > > **Further Discussion**
> > >
> > > Thank you for taking the time to review our responses and for your initial feedback.
> > >
> > > While we respect your decision to maintain your original score, we would like to reiterate our belief that we have comprehensively addressed all the concerns raised. If there are any specific areas where you feel our response could be further strengthened, we would be grateful for the opportunity to address them.

---

> > > > ### Comment · Reviewer_QVMC · 2023-11-21
> > > > **Further clarification is needed!**
> > > >
> > > > Thank you, but I believe your response to my third question is quite general. I've read the 'General Response' and Appendix C, but I remain unconvinced. Could you perhaps provide more details on that (3. How effectively would the proposed method perform in capturing the dynamics of a multi-stable system with **a stable fixed point and a periodic attractor**, or possibly **with two periodic attractors**?)?

---

> ### Author Response · Authors · 2023-11-22
> **Additional experiment and discussion**
>
> Thank you for your prompt reply. We hope the additional experiment and discussion below address your remaining concerns.
>
> ## [Additional Experiment](https://imgur.com/a/UyfhsHa)
>
> As an additional experiment, we trained our model on the **[Damped Duffing Oscillator](https://imgur.com/a/G1Cfjw3)**, a multi-stable NLDS, **characterized by two distinct periodic attractors with centers at $(\pm 1, 0)$**. We hope this sytem serves as a sufficiently rich example to answer the reviewer's question -- if it does not, we are happy to consider the NLDS the reviewer suggests. The results empirically validate the effectiveness of periodic reencoding for modeling multi-attractor systems. For a more detailed explanation with theoretic insight kindly see the "Theoretic Clarification."
>
> We report the average mean squared error (MSE) over a 1000-step long horizon to be $\underline{\textbf{1.4 e-3}}$ and $\textbf{57.3 e-3}$ with and without Periodic Reencoding, respectively. The error/phase plots are accessible via [_this link_](https://imgur.com/a/UyfhsHa). We use a Koopman Autoencoder with linear decoder identical to those used in the paper, and a dataset of 50 trajectories of length 500. Periodic Reencoding again outperforms never/every-step reencoding methods. The rollouts are generated from initial points not seen during training.
>
>
> ## Theoretic Clarification
>
> First, consider the special case of a linear decoder $\psi(z) = Wz$. Suppose we try to model the above system *without reencoding*. After training the autoencoder this corresponds to a three step generative process: **1.** the encoder is used to infer an initial condition $z_0 = \phi(x_0)$, **2.** a linear dynamical system, $\dot z = Kz$ generates the solution $z(t) = z_0e^{Kt}$ in the latent space, and finally **3.** this solution is mapped back to the phase space with the linear decoder to produce $x(t) = Wz_0e^{Kt}$. This last expression corresponds to a linear projection of the solution to a linear dynamical system. **Recall that linear dynamical systems can only have a single fixed point (if $\mathbf{K}$ is full rank) or a hyperplane of fixed points given by the null-space of $K$ if it is rank-deficient.** A linear system lacks the capacity to represent dynamics containing distinct fixed points (such as the Duffing Oscillator), regardless of its dimensionality ("Nonlinear Systems" by Hassan K. Khalil is a good reference). Clearly, a linear projection, $W$ also lacks the capacity to change the topology of the attractor structure of the solution (a point will map to a point, and a hyper-plane will map to a hyper-plane). As shown in the phase plots generated without reencoding, over long horizons, solutions clearly diverge from the ground truth and fail to capture multi-attractor dynamics. A plausible explanation for this is described in Appendix C and Section 3.2 which we rephrase again: **without reencoding, the dynamics remain committed to the initial linear system determined by the encoder, corresponding to $z_0$**. When reencoding happens at $t=\Delta t$, the system is enabled to switch to between solutions, $x(t) = W \mathbf{\hat{z}}\_{\Delta t} e^{Kt}$, corresponding to the basin of attraction of possibly another fixed point — where $\mathbf{\hat{z}}_{\Delta t} = \phi \circ \psi(z_0 e^{K \Delta t})$, see notation in Figure 2. This is reflected in the results from the above paragraph, in that multi-attractor dynamics are better captured with reencoding.
>
> ---
>
> _We trust that the theoretical clarification along with the additional experiments address your remaining concern._

---

> > ### Comment · Reviewer_QVMC · 2023-11-22
> > **Many thanks!**
> >
> > I greatly appreciate the authors for their detailed response! My concerns have been addressed, and I will increase my score to 8.

---

### Author Response · Authors · 2023-11-16
**General Response to Reviewers**

We thank all the reviewers for your reviews and feedback. We have incorporated your suggestions into our paper and will continue refining it for the camera-ready version. We also plan to open-source our code in the near future, aiming to facilitate the adoption of our work within the community. Our responses to the reviewers are submitted early with the intention of fostering ample time for active discussion.

---

This general response highlights some of the points that perhaps were not clear in the original draft, crucial in conveying the core contributions of the paper.

Equations 8 and 9 from **Section 3.2** concisely illustrate the difference between modeling a NLDS with/without reencoding:

* When reencoding is used, the model (encoder/decoder/$\mathit{K}$) themselves act as NLDS, i.e. a feedback process in $x$ (Equation 9), and benefit from inductive biases this offers, including satisfying the IVP theorem by construction.
* When no reencoding is used, Equation 8 illustrates that the model is not a feedback process in $x$, but rather a linear feedback process only in $z$, and reconstructs the NLDS training trajectories by applying an open-loop non-linearity (decoder), pointwise, to reconstruct the training trajectories produced by the underlying NLDS. This maps _trajectories_ in $z$-space, to _curves_ in the $x$-space, which do not necessarily satisfy the IVP principle. The issue stems from the non-injective nature of the mappings into the $x$-space, especially when $dim(z) > dim(x)$.

Seemingly, a nonlinear function with enough capacity can represent any input-output relationship (Equation 8), however, modeling a NLDS with another, trainable NLDS (realized by reencoding) is an excellent (perhaps essential) inductive bias.

Indeed, the work by Lan and Mezic shows that multi-attractor NLDS can only be linearized within each basin of attraction. Two examples in the paper illustrate this concretely:

*  **Appendix C** describes a sparse Koopman autoencoder which learns to partition the input space into regions where the dynamics are approximately linear. Because the decoder is linear, it lacks the capacity to infer a new support when the dynamics switch between these linearized regions. This is one reason for “long term drift” `R-Fg8g` – over-extrapolation using a stale linear approximation instead of switching to a new linear approximation. This is why the “Duffing Oscillator” results look like a superposition of linear dynamical systems in Figure 4(a). Reencoding queries the encoder to infer a new sparse code with a new support. This is not “a noisy perturbation” `R-f4vn` but a property of the encoder – it implicitly learns to assign a constant support to regions of the state space where the dynamics are approximately linear. The distinct support for each region is _not "randomly"_ assigned, but must correspond to the appropriate linearized representations corresponding to each region of the state space.
* The Parabolic Attractor example illustrates the converse point: there are some NLDS which can be perfectly modeled without reencoding (Figure 3) – such systems can only have a single topologically distinct attractor. Indeed it is clear to see analytically that the features, $z$ in Equation 11, linearize the dynamics, without needing reencoding, and can be used to reconstruct $x$ with a linear decoder.

The analysis (Equations 8 & 9) as well as special examples (Figure 3 & Appendix C) analytically illustrate and provide intuition for why reencoding is necessary and can’t easily be replaced by, for example, architectural biases in the encoder/decoder. Our paper aims to illustrate an important point missed by other works on Koopman auto-encoders, and indeed any work that models dynamics in a latent space. _Periodic reencoding is a method that can enhance the performance of many approaches, and is not meant to compete with other innovations._

---

We have also added additional results in **Appendix E** of the revised submission, which include:
* Complementary error plots for D4RL tasks (**Figure 7**).
* Error plots for models trained without periodic reencoding at training time, unrolled under varying reencoding schemes (**Figure 8**).
* Experimental results on:
    * Deep Latent-to-Latent Mapping
    * Single-step horizon Training Objective
    * Imposing Various Structures on the Koopman Matrix

---

We will try our best to respond promptly to your additional questions. We thank the reviewers for their time and feedback.

---

### Author Response · Authors · 2023-11-22
**Awaiting Responses from Reviewers: FYpR, Fg8g, f4vn**

_The official public discussion phase is ending today._ We ask reviewers `R-FYpR`, `R-Fg8g`, and `R-f4vn` to please consider our general and specific responses. We believe that we have addressed all of your questions.

Thank you.

---

### Meta-Review · Area_Chair_NQrL · 2023-12-08

**Metareview:**

Reviewers have acknowledged the clarity and novelty of the manuscript, particularly highlighting the concept of periodic reencoding for long-term forecasting. Although there were initial concerns regarding the theoretical benefits and practical implementation, the authors' comprehensive responses and additional experiments significantly addressed these issues. The paper presents an interesting approach within the realm of forecasting in nonlinear dynamical systems. As a total, this paper is recommended for acceptance (poster).

**Justification For Why Not Higher Score:**

Although presents an interesting approach within the realm of forecasting in nonlinear dynamical systems, it seems to fall short of a major breakthrough, which is why an acceptance for this format, rather than a higher one.

**Justification For Why Not Lower Score:**

See above.

---

### Decision · Program_Chairs · 2024-01-16

Accept (poster)